# A Review of Posteromedial Lesions of the Chest Wall: What Should a Chest Radiologist Know?

**DOI:** 10.3390/diagnostics12020301

**Published:** 2022-01-25

**Authors:** Sara Haseli, Bahar Mansoori, Mehrzad Shafiei, Firoozeh Shomal Zadeh, Hamid Chalian, Parisa Khoshpouri, David Yousem, Majid Chalian

**Affiliations:** 1Department of Radiology, Division of Musculoskeletal Imaging and Intervention, University of Washington, Seattle, WA 98105, USA; sarahaseli@gmail.com (S.H.); mshafie@uw.edu (M.S.); shomal@uw.edu (F.S.Z.); khoshpouriparisa@gmail.com (P.K.); 2Department of Radiology, Division of Abdominal Imaging, University of Washington, Seattle, WA 98105, USA; mansoori@uw.edu; 3Department of Radiology, Division of Cardiothoracic Imaging, University of Washington, Seattle, WA 98105, USA; Hamid.Chalian@hsc.utah.edu; 4Russell H. Morgan Department of Radiology and Radiological Sciences, Division of Neuroradiology, Johns Hopkins Medical Center, Baltimore, MD 21287, USA; dyousem1@jhu.edu

**Keywords:** chest wall, posteromedial, lesion, imaging, benign, malignant

## Abstract

A heterogeneous group of tumors can affect the posteromedial chest wall. They form diverse groups of benign and malignant (primary or secondary) pathologies that can arise from different chest wall structures, i.e., fat, muscular, vascular, osseous, or neurogenic tissues. Chest radiography is very nonspecific for the characterization of chest wall lesions. The modality of choice for the initial assessment of the chest wall lesions is computed tomography (CT). More advanced cross-sectional modalities such as magnetic resonance imaging (MRI) and positron emission tomography (PET) with fluorodeoxyglucose are usually used for further characterization, staging, treatment response, and assessment of recurrence. A systematic approach based on age, clinical history, and radiologic findings is required for correct diagnosis. It is essential for radiologists to be familiar with the spectrum of lesions that might affect the posteromedial chest wall and their characteristic imaging features. Although the imaging findings of these tumors can be nonspecific, cross-sectional imaging helps to limit the differential diagnosis and determine the further diagnostic investigation (e.g., image-guided biopsy). Specific imaging findings, e.g., location, mineralization, enhancement pattern, and local invasion, occasionally allow a particular diagnosis. This article reviews the posteromedial chest wall anatomy and different pathologies. We provide a combination of location and imaging features of each pathology. We will also explore the role of imaging and its strengths and limitations for diagnosing posteromedial chest wall lesions.

## 1. Introduction

Chest wall tumors are uncommon causes of thoracic neoplasms, which are less common than soft tissue or bony neoplasms elsewhere. Unfamiliarity with the complex posteromedial chest wall anatomy and radiologic features of related neoplasms is a diagnostic dilemma for radiologists [1]. These tumors are heterogeneous with nonspecific clinical manifestations and different imaging characteristics, which make their diagnosis challenging. Either a benign or malignant nature and primary or secondary origin are probable [2,3,4]. Primary chest wall neoplasms originate from chest wall structures, e.g., bony thorax, cartilage, muscle, fat, blood vessels, and nerve sheet [3,5]. Secondary chest wall neoplasms include direct invasion from adjacent malignancies (lung or breast carcinomas) or distant metastasis [2]. 

The posteromedial aspect of the chest wall has complex anatomy due to the presence of intercostal nerves, sympathetic chain, and vascular structures. Many neoplasms originate from these structures [6]. Some of them may be almost exclusive to this location. Neurogenic tumors are more commonly arising from the posteromedial chest wall as they originate from autonomic ganglia, paraganglia, or nerve sheets. So, they account for the majority of lesions found in the posterior mediastinum and chest wall [7]. Many of these lesions have specific imaging characteristics that help make precise diagnoses and avoid invasive sampling. In other conditions with nonspecific imaging appearance, cross-sectional imaging plays an essential role in limiting the differential diagnosis and defining the further investigation, e.g., imaging-guided biopsy. So, it is crucial for radiologists to be familiar with these diverse group of lesions and their imaging characteristics [4,8]. 

Previous studies mostly focused on the assessment of malignant lesions of the chest wall. None of them specifically evaluated the lesions of the posteromedial chest wall [9,10,11]. Only one review article investigated the paravertebral masses in the thoracic boundary. This study categorized lesions into neurogenic tumors, non-neurologic tumors, and non-neoplastic masses [6]. To the best of our knowledge, our review is the only one focusing on the posteromedial aspect of the chest wall, addressing nearly all of the lesions that could be found in this anatomic location. This article reviews the posteromedial chest wall anatomy and different pathologies. We illustrated the imaging features of each lesion, e.g., the location, presence of calcification, adjacent bone destruction, the pattern of enhancement, and appearance on magnetic resonance imaging and positron emission tomography. We also explored the role of imaging and its strengths and limitations for diagnosing posteromedial chest wall lesions.

## 2. Posteromedial Chest Wall Anatomy

In this article, we focus on the posteromedial segment of the thoracic wall. The medial aspect of the posterior chest wall consists of multiple components detailed below (Figure 1):Osseous/cartilaginous parts: 12 thoracic spine vertebrae, 12 ribs, and intervertebral discs.Muscles: Intercostal muscles (external, internal, and innermost), subcostalis, and transverse thoracic.Nerves: intercostal nerves, dorsal root ganglions, and sympathetic trunk.Vascular tissues: Intercostal vessels feed above components.Subcutaneous fat: beneath the superficial fascia and builds the padding for underlying muscles and bones.Superficial fascia and skin: acting as protecting layers [5,12].

## 3. Classification of Posteromedial Chest Wall Lesions

Chest wall neoplasms are heterogonous lesions, and there is a lack of universally accepted classification. So, these neoplasms are usually classified according to the tissue of origin Table 1.

## 4. Role of Imaging

Chest radiography is usually the first imaging modality performed when chest wall lesions are clinically suspected, although it can provide some nonspecific information [4]. More specific cross-sectional modalities such as computed tomography (CT) and magnetic resonance imaging (MRI) are warranted for better tissue characterization and assessment of lesion extension [2]. Obtaining high-resolution images as well as a short acquisition time makes CT the modality of choice for the initial evaluation of chest wall lesions. Although CT is more precise in assessing bone lesions, MRI has a superior contrast resolution, revealing more details regarding tissue characterization and tumor extension [2,4,6]. 18F-Fluorodeoxyglucose positron emission tomography/computed tomography (18F-FDG PET/CT) is another complementary modality beneficial in initial staging, evaluation of response to treatment, and tumor recurrence [2,13].

On the other hand, recent advances in deep learning and artificial intelligence (AI) provide the ability of automatic classification, disease detection, and segmentation. Chest radiography and CT scan are excellent candidates for developing deep learning algorithms [14]. AI has the potential to detect visual information and perform quantitative analyses. Besides, radiomics can be used to characterize the benign or malignant nature of a lesion and predict the prognosis and probability of response to treatment of the malignant lesions (Figure 2) [14,15].

## 5. Malignant Bone Tumors

Radiologic characteristics of malignant bone tumors are summarized in Table 2.

### 5.1. Osteosarcoma

Chest wall osteosarcoma is a high-grade tumor accounting for 10–15% of primary chest wall malignancies, which can arise from rib or scapula with an extrapleural component. It has a poor prognosis as the lung and lymph nodes metastases are more frequent than extremities osteosarcoma [1,2].

A sclerotic lesion with higher central calcification is the most common appearance of osteosarcoma on radiography. Cortical destruction, expansile remodeling, and periosteal reaction are other common findings. CT can show soft-tissue destruction with variable types of calcifications such as cloudy, ivory-like, or dense (Figure 3) [1,2]. On MRI, the soft tissue component of osteosarcoma shows low to intermediate intensity on T1-weighted and hyperintensity on T2-weighted images. Foci of matrix mineralization have low signal intensity on both T1-weighted and T2-weighted images. Osteosarcoma demonstrates heterogeneous postcontrast enhancement. Invasion of deeper structures, pathologic fracture, and spinal canal invasion also have been described [1,2]. 

### 5.2. Ewing Sarcoma

Chest wall Ewing sarcoma is the most common primary chest wall tumor in children, typically arising from ribs or, less frequently, scapula, clavicle, and sternum. Painful chest wall mass with fever and malaise are the most common presentations [16].

On CT, Ewing sarcoma appears as a destructive paraspinal soft tissue mass with internal necrosis and hemorrhage. Calcification is rarely reported, and the soft tissue component is usually larger than the osseous component [11]. It appears iso to hyperintense on T1-weighted and heterogeneous to hyperintense on T2-weighted MRI sequences. Intense homogenous or heterogeneous post-contrast enhancement is expected. Extraosseous Ewing sarcoma, usually seen in older patients, manifests as a large non-calcified mass within the paravertebral location (Figure 4) [17]. 18F-FDG PET/CT is a valuable complementary tool enabling accurate identification of adjacent invasion and distant metastasis [18,19].

### 5.3. Chondrosarcoma

Chondrosarcoma is the most common primary neoplasm of the chest wall accounting for 30% of malignant lesions [2]. It develops as either primary or malignant degeneration of preexisting benign lesions. It usually occurs between the 4th to 7th decades with a male predilection [1]. 

The typical CT appearance is a well-defined destructive mass with a mixture of soft tissue and mineralized components with stippled, dense, flocculent, rings, or arcs patterns. The invasion of adjacent structures has also been described [17]. On MR imaging, background cartilage shows iso- to hypointensity on T1-weighted and hyperintensity on T2-weighted images. The area of mineralization is low signal on both T1-weighted and T2-weighted images. Heterogeneous post-contrast enhancement is seen especially at the periphery of lesions with linear or septa-like patterns [17]. It is supposed that 18F-FDG PET/CT can play a complementary role in characterizing the chondrosarcoma from chondroma (Figure 5) [20]. 

## 6. Malignant Plasma Cell Tumors

### 6.1. Multiple Myeloma

Multiple myeloma (MM) is an infiltrative bone marrow disorder and concurrent bone lesion predominantly involving the axial skeleton, including vertebral bodies, skull, pelvis, and ribs [21].

Multiple myeloma radiologic features are osteolytic lesion with endosteal scalloping, diffuse osteopenia, multiple small lesions with mottled appearance, and osteoporotic fracture. In contrast to bone metastasis, the sclerotic halo is absent in MM, explaining the lower accuracy of scintigraphy compared to skeletal survey. Whole-body low dose CT was recommended as the initial survey for diagnosing osteolytic lesions of MM, which improved identification of extraosseous involvement and cortical disruption. STIR and T2-weighted imaging are the most sensitive sequences for depicting marrow signal changes. The T1-weighted sequence is useful for the evaluation of marrow infiltration [21,22].

#### Solitary Plasmacytoma of the Bone

Solitary plasmacytoma (SBP) of bone is an uncommon plasma cell neoplasm with localized bony growth. There are few case reports on solitary plasmacytoma of the rib [23,24]. 

Its radiologic appearance varies from purely non-expansile osteolytic to multicystic mass with bony expansion. CT may reveal extrapleural mass with a well-circumscribed margin and “soap bubble” appearance in advanced cases (Figure 6a,b). MRI shows T1-hypointensity and T2-hyperintensity (Figure 6c) [16,25]. SBP tends to show metabolic activity on 18F-FDG PET/CT, which seems to be a risk of multiple myeloma transformation (Figure 6d,e) [26]. 

## 7. Benign Bone Tumors

Radiologic characteristics of benign bone tumors are summarized in Table 2.

### 7.1. Aneurysmal Bone Cyst

An aneurysmal bone cyst (ABC) is an uncommon benign bone tumor that consists of multiple blood-filled cysts, which can present as a primary tumor or as secondary changes of other bone tumors [1,16,27]. The most common location of chest wall ABCs is vertebral bodies with extension through adjacent soft tissue structures [27]. 

ABC’s imaging appearance on radiography and CT is a well-defined expansile osteolytic lesion with thin marginal sclerosis, typical fluid-fluid level, and internal septation [1]. T1-weighted hyperintensity might be found secondary to the subacute timeline of internal hemorrhage (Figure 7) [16,27].

### 7.2. Fibrous Dysplasia

Fibrous dysplasia (FD) is a developmental bone lesion caused by immature bone and marrow fibrous tissue replacement. It affects patients during the first and second decades of life. Rib is the most commonly affected site [1,16]. 

Fibrous dysplasia presents as a well-defined intramedullary osteolytic lesion with fusiform bony expansion and endosteal scalloping with preservation of cortical contour (Figure 8a). Increased trabeculation, thickened cortex, and “Ground glass” appearance caused by amorphous woven bone formation are other imaging findings (Figure 8c). FD has typical low intensity on T1-weighted and variable low to high intensity on T2-weighted MRI sequences depending on varying amounts of fibrous tissue (Figure 8b) [16,28]. FD metabolic activity on 18F-FDG PET/CT ranges from normal to intense, and it depends on the number of proliferating fibroblasts [29].

### 7.3. Giant Cell Tumor

Chest wall giant cell tumor (GCT) originates from the subchondral region of tubular or flat bones, e.g., clavicle, ribs, sternum, and vertebrae. It usually affects patients during the third or fourth decade of age, with a female predilection. Thoracic vertebrae is the most common location for spinal GCT after sacrum [8,30]. 

On CT, GCTs manifest as osteolytic lesions, bone expansion with cortical thinning, heterogeneous soft-tissue attenuation, and area of hemorrhage or necrosis with no internal calcification (Figure 9a,b) [30]. MRI typically reveals low to intermediate intensity on both T1-weighted and T2-weighted sequences, representing an abundant amount of hemosiderin and collagen deposition. Fluid-level occurrence is less frequent than in ABC (Figure 9c,d) [8,31]. 18F-FDG PET/CT may cause misdiagnosis of the GCT as a high-grade osseous sarcoma [30]. 

### 7.4. Enchondroma

Enchondroma is the most common benign rib neoplasm after fibrous dysplasia, accounting for 15–20% of benign rib tumors. It typically arises from the costochondral or costovertebral junction due to its cartilaginous nature [32]. 

CT reveals focally expansile well-demarcated osteolytic lesion with or without cortical bulging. MRI helps depict enchondroma, especially when matrix calcification is vague on CT. Enchondroma has pronounced hypointensity on T1-weighted images as opposed to adjacent hypersignal fatty bone marrow. Hyaline cartilaginous content [rich in water] results in hyperintensity on T2-weighted images, and internal calcification foci produce hypointensity on all sequences [8,32].

### 7.5. Chondromyxoid Fibroma

Chondromyxoid fibroma is a rare benign cartilaginous tumor that contains various proportions of fibrous, myxomatous, and chondroid components [1,6]. It infrequently occurs within the chest wall from the scapula, spine, or ribs [33].

Cortical expansion with lobulated border, abundant peripheral sclerosis, and rarely internal calcification are the main radiologic appearances of chondromyxoid fibroma on radiograph and CT [3,33]. Cortical expansion with lobulated border, abundant peripheral sclerosis, and rarely internal calcification are the main radiologic appearances of chondromyxoid fibroma on radiograph and CT [3,33]. On MRI, varying degrees of signal intensity can be identified. Isointensity on T1-weighted and intermediate to high intensity on T2-weighted images have been reported. Peripheral hypointense rim seen on both T1-weighted and T2-weighted images reflects the sclerotic rim. The absence of diffusion restriction and diffuse moderate to intense contrast enhancement is noticeable (Figure 10) [3,33]. The absence of diffusion restriction and diffuse moderate to intense contrast enhancement is noticeable (Figure 10) [3,33].

### 7.6. Chondroblastoma

Chondroblastoma is a rare, relatively benign bone neoplasm with a characteristic intercellular cartilaginous matrix and internal foci of calcification. It is usually found in the epi-metaphyseal regions of long bones. However, few cases of rib involvement have been reported [1,34]. It is frequently associated with cortical destruction and periosteal bone formation resembling malignant bone tumors [1,34,35].

The main radiologic appearance of chondroblastoma on radiograph and CT is an oval or round well-circumscribed lesion with internal matrix mineralization [1,34]. It appears homogenously hypointense on T1-weighted and heterogeneous on T2-weighted MRI sequences [36].

### 7.7. Paget’s Disease of the Rib

Paget’s disease is a chronic bone disorder characterized by abnormal osseous remodeling with three phases: lytic, mixed lytic and blastic, and sclerotic. It occurs in patients older than 40 years old with a male predilection. Rib involvement happens approximately in 1–4% of cases [35].

Imaging findings vary among different stages of the disease. The initial lytic active phase display osteolysis with no marginal sclerosis. While late blastic phase reveals cortical thickening and trabecular coarsening with bony enlargement. CT scan is useful in better delineation of the classic Paget’s disease triad: osseous expansion, cortical thickening, and trabecular coarsening (Figure 11) [35,37]. Fat-like signal intensity is the most common pattern of the Paget disease seen on MRI, indicating long-lasting disease. Other stage-specific MRI findings are explained in Table 2 [35,37]. The relationship between 18-F FDG uptake and Paget’s disease activity is still controversial, but a mild and diffuse pattern of FDG uptake is beneficial to differentiate it from bone metastasis [38].

## 8. Spondylodiskitis

Spondylodiskitis is an infectious process that initially affects the anterior portion of the vertebral bodies and then spreads to the adjacent intervertebral disk via medullary spaces [39]. Staphylococcus aureus is the most common cause of pyogenic spondylodiskitis, which commonly presents as a single-level lumbar involvement of two vertebral bodies and intervertebral disk. Tuberculosis spondylodiskitis, the most common non-pyogenic spine infection, more commonly involves the thoracic spine [39,40].

### 8.1. Pyogenic Spondylodiskitis

Pyogenic spondylodiskitis presents with loss of vertebral end plate definition and marrow edema. It displays hypointensity on T1-weighted images and hyperintensity on T2-weighted and STIR images. Various types of disk post-contrast enhancement (e.g., homogenous, patchy, and peripheral) may be detected [39]. Abscess or phlegmons demonstrate heterogeneous mixed signal intensity on both T1-weighted and T2-weighted images, with probable spinal cord compression. Rim-like or diffuse post-contrast enhancement are usually seen within these soft tissues. Diffusion weighted imaging (DWI) is valuable for differentiating the abscess from other paravertebral lesions (Figure 12) [39,41].

### 8.2. Tuberculosis Spondylodiskitis

Tuberculosis spondylodiskitis has a more gradual and chronic clinical course, which leads to multi-level involvement and paravertebral cold abscess formation with well-circumscribed thin wall. Subligamentous spread of infection to adjacent vertebral levels, relative preservation of intervertebral disk, and kyphotic angulation (gibbous deformity) are other imaging findings. CT scan is more sensitive in delineating calcification within paravertebral cold abscess, end plate erosion, and bony fragment visualization (Figure 13) [39,42].

## 9. Soft-Tissue Tumors and Tumor-Like Lesions

### 9.1. Primary Neurogenic Lesions

Neurogenic tumors of the chest wall can arise from the intercostal nerve, spinal nerve roots, and even from the distal branch of the brachial plexus. They consist of benign and malignant groups, including Neurofibroma, Schwannoma, and malignant peripheral nerve sheet tumors (Table 3).

#### 9.1.1. Schwannoma

Schwannoma is an encapsulated slow-growing peripheral nerve sheet neoplasm typically occurring in patients between 20–50 years old [6,16]. Chest wall schwannomas arise from spinal nerve roots with a dumbbell shape appearance and extend through the course of intercostal nerves, paravertebral region, or spinal canal [7,43].

Schwannoma presents a well-defined homogenous mass on CT scan with attenuation similar or less than muscle. The “Fat-split” sign caused by adjacent surrounding fat is indicative of its non-infiltrating growing pattern. It also shows remarkable post-contrast enhancement except for areas of necrosis or cystic changes (Figure 14). On MR images, it has intensity equal to or slightly more than muscle on T1-weighted and marked hyperintensity on T2-weighted images. Scalloping or bony erosions might be the only radiographic manifestations reflecting its benign nature [6,7,16,43].

#### 9.1.2. Neurofibroma

Neurofibroma is another slow-growing peripheral nerve sheet neoplasm that affects patients in their 20s to 30s with equal male and female prevalence. Localized Neurofibroma, which includes approximately 90% of cases, is not typically associated with neurofibromatosis type 1 (NF1). However, the majority of cases with plexiform type have underlying NF1 [7,16].

The main CT findings are well-circumscribed mass with smooth margin, soft tissue attenuation, possible internal calcifications, and adjacent rib erosion. Neural foraminal widening secondary to tumor extension can be accurately identified on multidetector CT (Figure 15). “Target sign” appears on both T2-weighted and gadolinium-enhanced MR images. It is related to the peripheral abundant stromal matrix surrounding the high cellular center, presenting as hyperintense rim and hypointense center, respectively (Figure 16 and Figure 17) [7,16,43].

#### 9.1.3. Neuroblastoma

Thoracic Neuroblastoma is a non-encapsulated tumor that arises from extra-adrenal sympathetic ganglia. The mediastinum is the second most common tumor location after the abdomen that has a better prognosis than other sites.

Neuroblastoma appears as an ill-defined paravertebral soft tissue mass on a CT scan with heterogeneous attenuation caused by hemorrhage, necrosis, or cystic degeneration. Internal calcification is seen at least in 30% of cases [7,16]. MRI shows irregular margin with possible local invasion to the spinal canal, presenting T1-hypointensity and T2-hyperintensity with heterogeneous enhancement. Calcification has a signal void in all sequences (Figure 18 and Figure 19). It is reported that tumors with higher metabolic activity on 18F-FDG PET/CT have lower overall survival [44,45]. Metaiodobenzylguanidine labeled as 123I (MIBG) is highly sensitive for detecting catecholamine-producing tumors like neuroblastoma [7,16].

#### 9.1.4. Ganglioneuroma

Ganglioneuromas are differentiated slow-growing neurogenic tumors originating from sympathetic ganglia that affect young patients. They appear as a paravertebral oval mass with a smooth border and vertical orientation following the sympathetic chain direction. Posterior mediastinum is the most common site of involvement of thoracic ganglioneuroma.

It has homogenous or heterogeneous attenuation on CT images with probable internal calcification (speckled, fine, or coarse). MR imaging displays lesions with intermediate signal intensity on both T1-weighted and T2-weighted images with curvilinear or nodular low signal bands, which form the whorled appearance (Figure 20) [7,16,43].

### 9.2. Lateral Meningocele

Lateral thoracic Meningocele is a rare condition defined as herniation of meninges through the vertebral column defect or enlarged neural foramina. It can be unilateral or bilateral and is usually associated with neurofibromatosis type 1 (Table 4). It is most common during the 4th to 5th decades of age, with female predominance [46].

Lateral Meningocele presents as a well-circumscribed paravertebral mass with similar attenuation to CSF. CT myelography reveals ipsilateral neural foramina enlargement communicating with subarachnoid space. This is a key differentiation feature from Neurofibroma [45]. On MR imaging, lateral Meningocele has T1-hypointensity and T2-hyperintensity with no post-contrast enhancement identical to CSF (Figure 21) [25,46].

### 9.3. Pseudomeningocele

Pseudomeningocele or meningeal pseudocyst is an abnormal extradural CSF collection that communicates with the brain and spinal canal. It can be congenital (thoracolumbar), traumatic (cervical), or iatrogenic (laminectomy of the lumbar spine). Congenital Pseudomeningocele can be seen in Marfan syndrome or NF1 [25].

The Pseudomeningocele can be differentiated from Meningocele by lack of dura wrapping the collection. The absence of nerve roots within the CSF collection helps in identifying the brachial plexus Pseudomeningocele. On MR imaging, it has similar intensity to CSF with a lack of post-contrast enhancement (Figure 22) [25,47]. The Pseudomeningocele can be differentiated from Meningocele by lack of dura wrapping the collection. The absence of nerve roots within the CSF collection helps in identifying the brachial plexus Pseudomeningocele. On MR imaging, it has similar intensity to CSF with a lack of post-contrast enhancement (Figure 22) [25,47].

## 10. Lipomatosis Tumors

Radiologic characteristics of lipomatosis tumors are summarized in Table 3.

### 10.1. Lipoma

Chest wall fatty tumors are relatively common, and lipoma is the most frequent. It is a well-defined mesenchymal tumor arising from adipose tissue usually seen in patients between 50–70 years old. Most chest wall lipomas are located deeply, involve intramuscular or intermuscular layers, and show larger size with less distinct borders than superficial ones [1,8,16,50].

On multidetector CT scan, lipomas are homogenous and have similar attenuation to macroscopic fat with approximate −100 HU radiodensity (Figure 23); other non-adipose components such as calcification and septa might also be seen. On MR imaging, signal intensity is identical to subcutaneous fat on T1-weighted and T2-weighted images. It typically does not enhance gadolinium-enhanced MR images except for septa with less than 2 mm thickness [8,16,50,51].

### 10.2. Liposarcoma

Liposarcoma consists of lipoblasts with various differentiations. Well-differentiated type is the most common subtype with near 50–75% internal fat component. The less frequent subtypes are dedifferentiated, myxoid, pleomorphic, and mixed subtypes. Chest wall involvement is not common [1,2,8].

On multidetector CT, liposarcoma has higher attenuation than normal fat secondary to a mixture of fat and malignant cells. Necrosis and calcification are uncommon in well-differentiated subtypes in contrast to the myxoid subtype (Figure 24). On MR imaging, Myxoid Liposarcoma has hyperintensity on both T1-weighted and T2-weighted images. Dedifferentiated subtype should be suspected when an area of T2-hyperintensity and T1-hypointensity are identified within preexisting well-differentiated liposarcoma [16,50]. Septal thickening of 2 mm or more, older age, larger size, and nodular non-adipose components are features that help to categorize liposarcoma over lipoma [16,50]. 18F-FDG metabolic activity can predict the liposarcoma grading, although there are some overlapping features [1].

## 11. Pleural Diseases

### 11.1. Empyema Necessitance

Empyema necessitance is a chronic pleural space infection that can affect both immunocompromised and immunocompetent people. Empyema leakage to chest wall soft tissues manifests as an extrapleural collection of empyema. Its usual location is the anterior aspect of the chest wall. Mycobacterium Tuberculosis is the most prevalent pathogen, with Nocardia Asteroides, Actinomyces Israelii, Staphylococcus, Aspergillosis, and Blastomycosis spp. being less common [1,8,52,53].

On CT, a communication between the pleural and extrapleural collection is a pathognomonic finding of empyema necessitans (Figure 25). A peripheral rim of soft tissue inflammation and thickening, draining sinus tracts, and rib destruction with periosteal reaction are other radiologic findings. MRI is extremely helpful in detecting vertebral and spinal canal involvement if the posteromedial part of the chest wall is affected (Table 4) [54].

### 11.2. Asbestos-Related Pleural Diseases

Asbestos-induced conditions include non-neoplastic and neoplastic pleural and lung diseases ranging from pleural effusion, thickening, plaques to malignant mesothelioma, and lung cancer. Pleural plaques are the most common disease [48,49,55].

Paravertebral and anterior plaques are better delineated on CT scans than radiography (Figure 26). On MRI, pleural plaques are hypo to isointense to skeletal muscle on T1-weighted, and hypointense on T2-weighted images. These findings are representative of fibrosis and internal calcification (Table 4) [48,49,55].

### 11.3. Mesothelioma

Malignant mesothelioma is the most common primary tumor of the pleura, which is related to prior asbestos exposure with a relatively poor prognosis [56].

Multidetector CT effectively reveals the primary tumoral extension, lymphadenopathy, and extrathoracic metastasis (Figure 27 and Figure 28) [56]. Another CT finding is circumferential pleural thickening (most common finding) with extension along the fissures. Large or punctate osseous or cartilaginous differentiation is more in favor of malignant mesothelioma rather than linear calcification that usually occurs within asbestosis plaques. Dynamic contrast-enhanced computed tomography (DCE CT) enables measuring intratumoral capillary permeability and blood flow, which are beneficial in evaluating treatment response [56,57,58].

MR imaging and 18F-FDG PET/CT are useful in further evaluation of chest wall, diaphragm, and mediastinal invasion [56,57,58,59]. Malignant mesothelioma appears as unilateral hyperintense pleural effusion and pleural thickening with iso to slight hyperintensity to chest wall muscles on T1-weighted and moderate hyperintensity on T2-weighted images. Post-contrast enhancement is expected (Table 3). It is believed that higher metabolic activity on 18F-FDG PET/CT is associated with poor prognosis and shorter survival time [25,56,57,58].

## 12. Other Soft Tissue Lesions

### 12.1. Rhabdomyosarcoma

Rhabdomyosarcoma is a high-grade sarcoma that mostly affects children and usually appears before 40. Chest wall rhabdomyosarcomas are uncommon tumors with poor prognosis and usually manifest as rapidly growing mass with adjacent bone invasion and nerve compression [1,16].

CT scan demonstrates homogenous mass with no mineralized matrix invading adjacent bone and soft tissue structures (Figure 29) [16]. MRI is the modality of choice for better delineation of tissue characterization, tumor extension, and medullary involvement. It is isointense to muscle on T1-weighted and hyperintense on T2-weighted images. Typical homogenous or ring-like enhancements are expected. Non-enhanced areas of hypointensity are reflective of necrosis (Table 3) [1,16,60]. 18F-FDG PET/CT is valuable in the primary staging, restaging, prognosis, and therapeutic assessment [61].

### 12.2. Extramedullary Hematopoiesis

Extramedullary hematopoiesis is a compensatory marrow hyperplasia in conditions with insufficient red blood cell production. Paraspinal areas are less frequently involved than lymph nodes, spleen, and liver. It typically arises from posterior ribs in the lower thoracic and upper lumbar area as unilateral or bilateral masses with no adjacent rib destruction [62].

On CT, it presents as a heterogeneous mass with internal foci of fat and no calcification. Lesions with hematopoietic activity demonstrate heterogeneous intermediate intensity on both T1-weighted and T2-weighted sequences with subtle or no enhancement. However, lesions with no hematopoietic activity might show foci of hyperintensity on both T1-weighted and T2-weighted images due to fat component or hypointensity on both T1-weighted and T2-weighted images secondary to iron deposition (Table 4) (Figure 30) [62,63].

### 12.3. Extramedullary Plasmacytoma

Extramedullary plasmacytoma accounts for less than 3% of all plasma cell neoplasms and can occur in any location. It presents as soft tissue masses with nonspecific imaging manifestations [64].

Larger lesions show aggressive behavior such as infiltration and destruction of adjacent soft tissue and encasement of vascular structures. It typically presents as isointense compared to muscle on T1-weighted images and iso to hyperintensity on T2-weighted images [64,65].

## 13. Conclusions

Chest wall neoplasms are a group of heterogeneous lesions, and the posteromedial chest wall is a source of different pathologies due to its complex anatomy. Many of these pathologies can be differentiated by imaging. A comprehensive systematic approach with varying imaging modalities is needed to identify the correct diagnosis or limit the differential diagnosis and determine the appropriate further investigation. This article illustrates the various posteromedial chest wall pathologies and their imaging features. Ill-defined border, heterogeneous enhancement, and local invasion are more suggestive of a malignant lesion. In contrast, well-defined borders and the absence of local invasion or distant metastasis favor benign nature. The pattern of mineralization helps in the differentiation of osseous/cartilaginous neoplasm from other neoplasms. We also explored key imaging features as well as strengths and limitations of each imaging modality. Therefore, the familiarity of radiologists with the imaging features of posteromedial chest wall lesions is crucial and can avoid unnecessary invasive procedures. Future investigation is required using quantitative novel imaging modalities to increase the diagnostic accuracy of radiology. Furthermore, improvement in deep learning and radiomics may increase patients’ benefit from reduced need for biopsy and individualized treatment options.

## Figures and Tables

**Figure 1 diagnostics-12-00301-f001:**
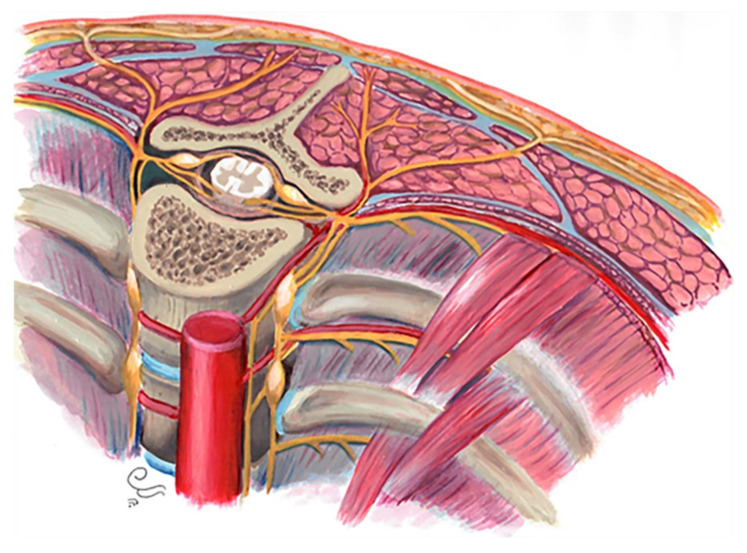
Posteromedial thorax anatomy, consisting of multiple components: skeletal components, muscles, nerves, ligaments, subcutaneous fat, fascia, skin, and vascular feeding tissues.

**Figure 2 diagnostics-12-00301-f002:**
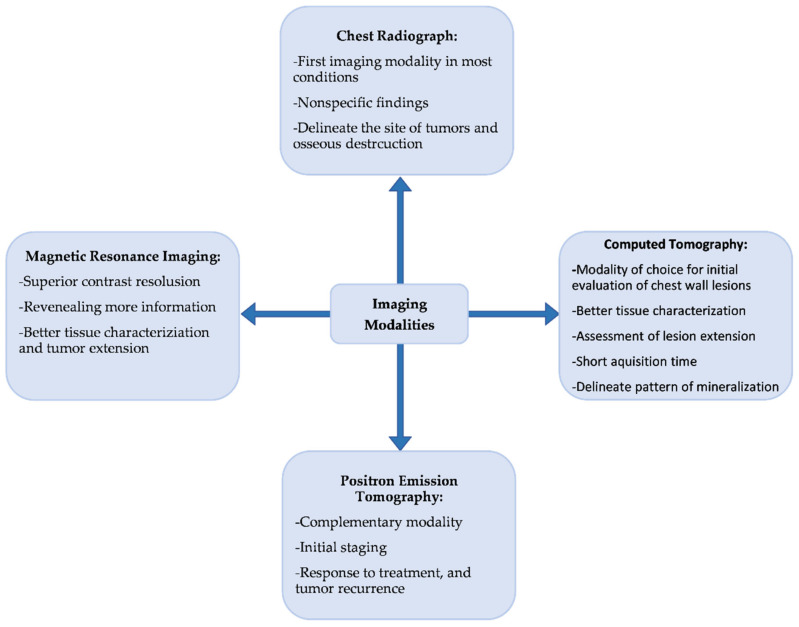
Tree graph of the most commonly used modalities for posteromedial chest wall lesions.

**Figure 3 diagnostics-12-00301-f003:**
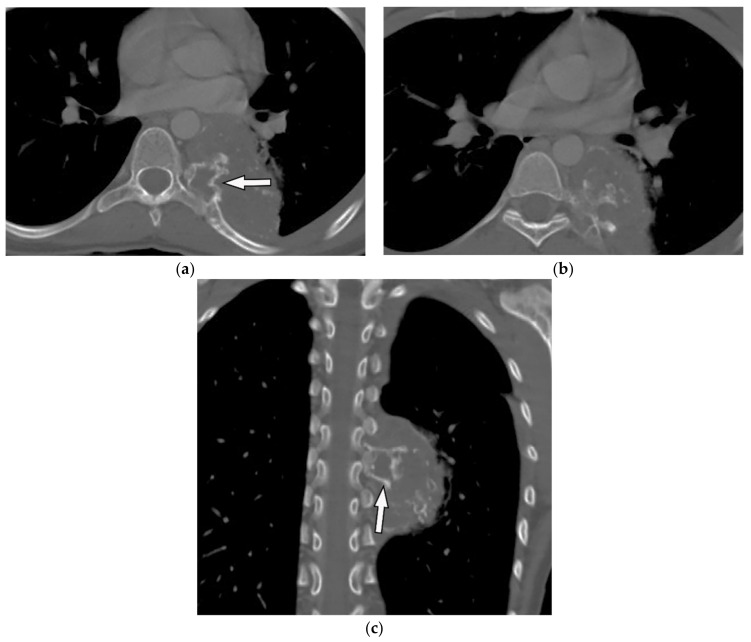
An 18-year-old woman with vague chest pain. The axial plane CT in the bone window obtained at the level of four chambers shows large destructive soft tissue mass within the posteromedial aspect of the chest wall on the left side with internal ossification that has a “sunburst” appearance (arrow) (**a**,**b**). The coronal view also shows the same large destructive soft tissue mass with internal calcification (arrow), which is denser centrally (**c**).

**Figure 4 diagnostics-12-00301-f004:**
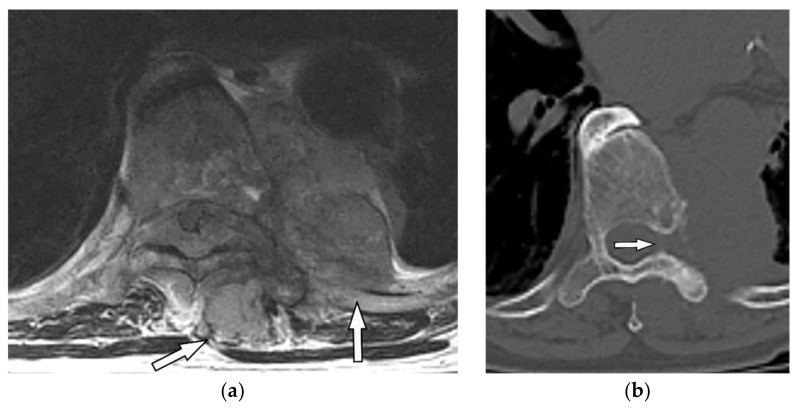
A 28-year-old man with bilateral leg weakness. The axial non-contrast-enhanced CT in the bone window (**a**) demonstrates a destructive mass in the posteromedial aspect of the chest wall on the left side, with a prominent soft tissue component with the left lateral vertebral body, pedicle, and lamina destruction (arrow). Axial T1W (**b**) MRI shows an inhomogeneous mass with iso to hyperintense signal intensity to paraspinal muscle with extension and invasion to left lateral recess, adjacent chest wall muscles, and rib (arrow).

**Figure 5 diagnostics-12-00301-f005:**
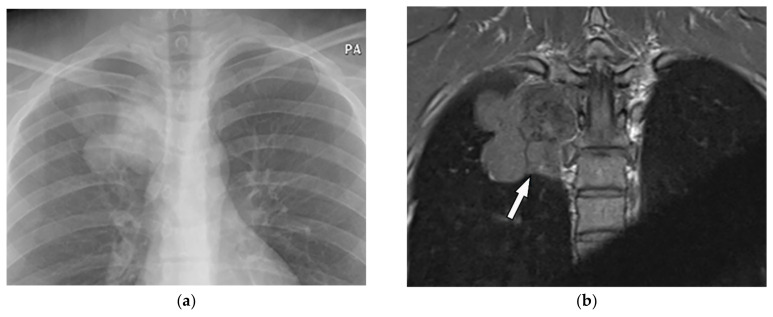
A 35-year-old man with chronic vague chest pain. (**a**) Frontal chest radiograph shows a right-sided lobulated mass with a cervicothoracic sign indicating the retro mediastinal location of the lesions. (**b**) Coronal T1W image shows lobulated paraspinal mass isointense to muscle (arrow). (**c**) Axial T2W fat-saturated image demonstrates heterogeneous high signal intensity with lobulated margin and internal foci of low signal intensity representing calcifications seen on CT (not shown). There is no neural foraminal extension.

**Figure 6 diagnostics-12-00301-f006:**
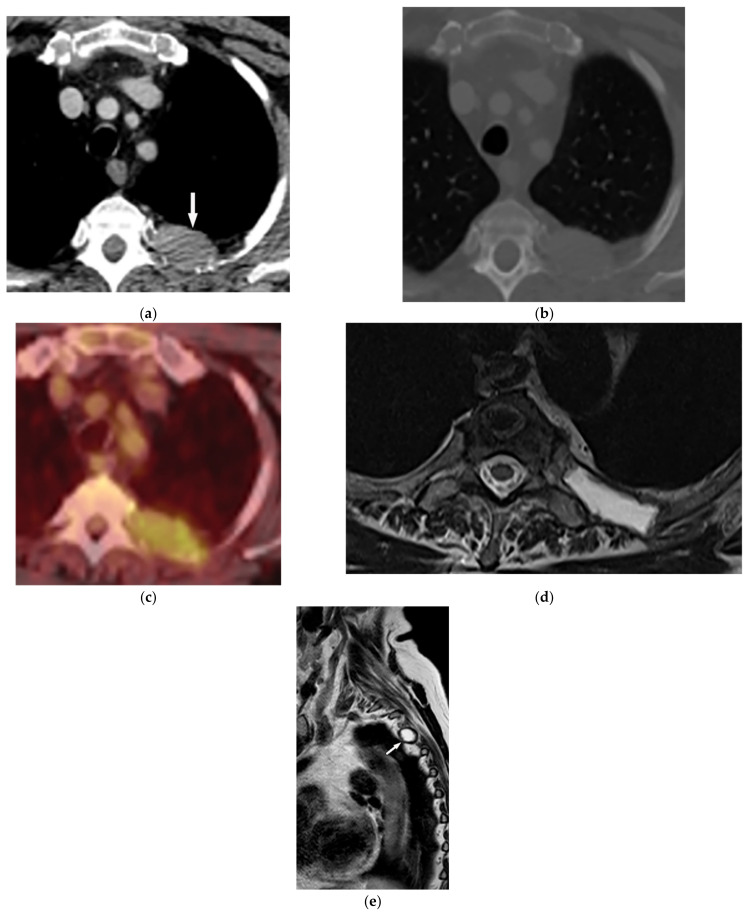
A 63-year-old woman with incidentally detected mass at the left posteromedial chest wall. Axial chest CT with mediastinal (**a**) and bone (**b**) window show a well-defined expansile mass (arrow) with the destruction of the adjacent rib. There is no internal calcification. No neural foraminal extension was seen (not shown). (**c**) T2W images of another patient with the same pathology show mildly expansile marrow replacing lesions at proximal posterior rib with high T2 signal intensity without cortical disruption or soft tissue mass. Increased metabolic activity is present on 18F-FDG PET/CT (**d**) Axial and (**e**) sagittal (arrow).

**Figure 7 diagnostics-12-00301-f007:**
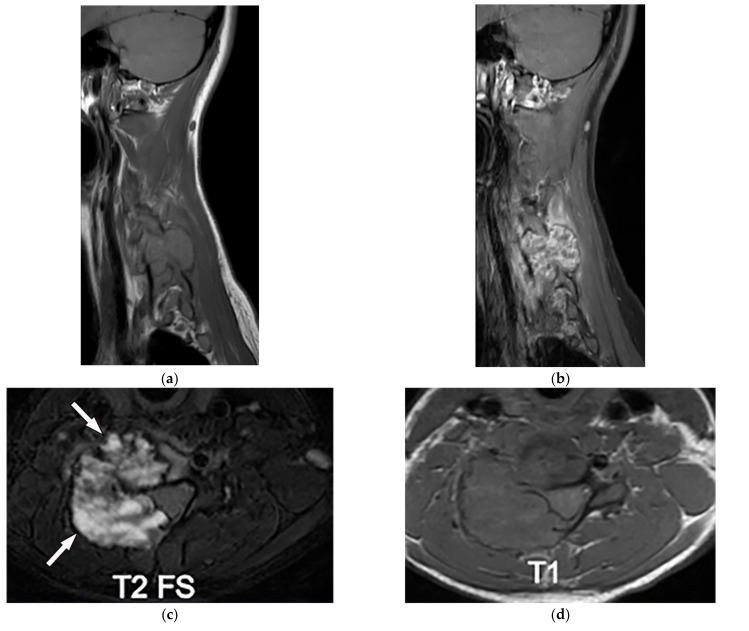
A 14-year-old boy with neck stiffness and scoliosis. The sagittal pre- and post-contrast T1WFS (**a**,**b**) demonstrate hypointense destructive spine lesion with low signal rim and heterogeneous enhancement of solid component. Axial images (**c**–**e**) show an expansile multiloculated mass (arrow) with the destruction of the vertebral body, spinous process, and lamina of the cervical spine. Fluid-fluid level and variable signal intensity caused by various-aged hemorrhage also were seen on T2WFs. It extends through the right lateral recess with canal stenosis. Slight hyperintensity on T1W is secondary to intratumoral hemorrhage. After injection of contrast, septal enhancements are seen.

**Figure 8 diagnostics-12-00301-f008:**
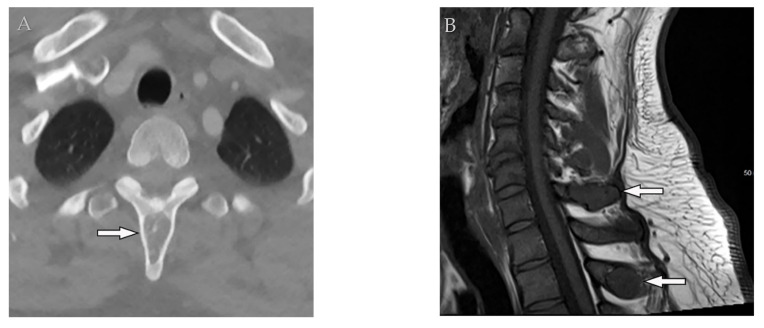
A 24-year-old man with multiple bone lesions. (**A**) The axial non-contrast CT in bone window demonstrates expansile lytic bone lesion with intact cortex within the spinous process of thoracic vertebrae (arrow); no periosteal reaction, cortical disruption, or soft tissue mass was found. (**B**) Sagittal T1W image shows expansile bone lesions with hyposignal intensity and intact cortex within the spinous process of multiple cervical spines (arrow). (**C**) Frontal image of pelvis shows bilateral expansile lytic bone lesions with bubbly appearance and typical shepherd’s crook deformity in the left side.

**Figure 9 diagnostics-12-00301-f009:**
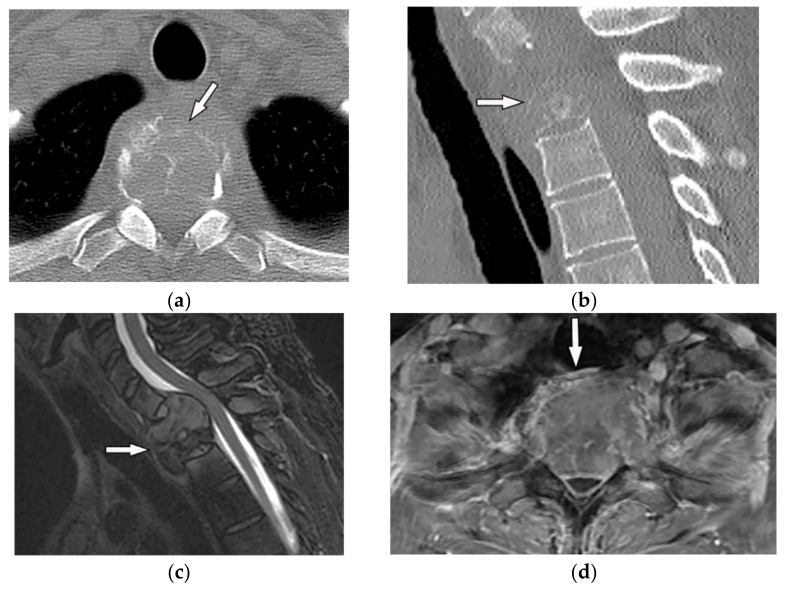
A 52-year-old man with bilateral leg weakness and paresthesia. The axial non-contrast CT in bone window demonstrates expansile osteolytic bone lesion with destruction and collapse of the vertebral body with pressure effect over the spinal cord and paravertebral soft tissue; the edge of vertebral body cortex disappeared (arrow) (**a**,**b**). Sagittal T2W shows heterogeneous iso signal mass with the collapse of the T2 vertebral body with compression of the dura and spinal cord (arrow); anterior soft tissue mass is also shown. The Intervertebral disc was intact. The visible internal hypointense lines are caused by thickened trabecular or hemosiderin deposition (**c**). T1W after injection of gadolinium shows heterogeneous marked enhancement of the tumor (arrow) (**d**).

**Figure 10 diagnostics-12-00301-f010:**
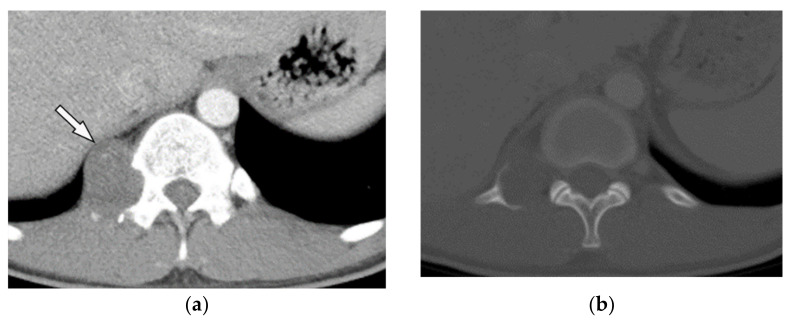
A 26-year-old man with right chest pain. The axial contrast-enhanced CT in soft tissue (**a**) and bone window (**b**) demonstrate well-defined oval eccentric lytic bone lesion within the posteromedial aspect of the chest wall on the right side (arrow) without intracanal extension or periosteal reaction; adjacent focal vertebral body scalloping is also shown. (**c**,**d**) Axial T1W and T2W show low signal intensity in T1W and intermediate to high intensity in T2W with the peripheral hypointense rim due to the sclerotic rim (arrow).

**Figure 11 diagnostics-12-00301-f011:**
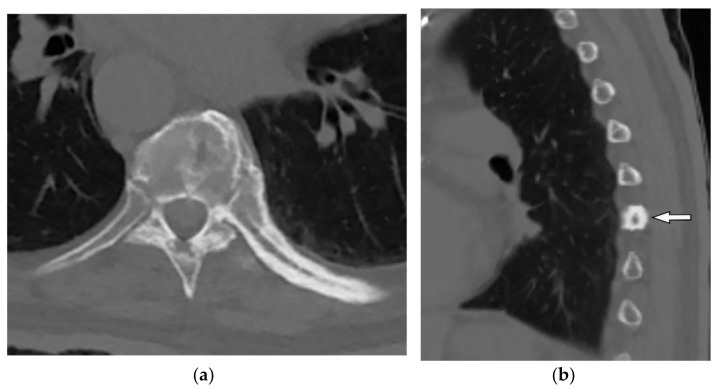
A 53-year-old woman with localized chest wall tenderness. The axial non-contrast CT in the bone window (**a**) and sagittal (**b**) images show enlarged vertebral body associated with trabecular coarsening, osseous expansion, and thickened cortex of right eighth rib (arrow).

**Figure 12 diagnostics-12-00301-f012:**
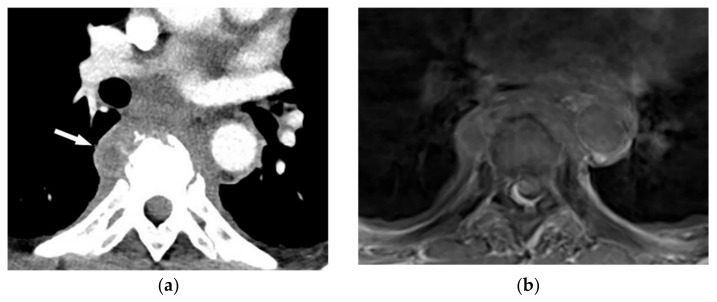
A 71-year-old man with back pain and fever. The contrast-enhanced CT in axial plane (**a**) demonstrates para-spinal soft tissue mass (arrow) with adjacent vertebral body cortical destruction. Axial pre-contrast (**b**) and post-contrast T1WFS (**c**) images at the level of T6-T7 show hypo to isointense paravertebral soft tissue mass with peripheral rim enhancement (arrow) after injection of gadolinium suggestive for paravertebral abscess formation. Sagittal STIR (**d**) image shows adjacent subchondral bone marrow edema as hypersignal intensity. Posterior elements are spared with normal signal intensity. Aspiration was performed and culture was compatible for brucellosis.

**Figure 13 diagnostics-12-00301-f013:**
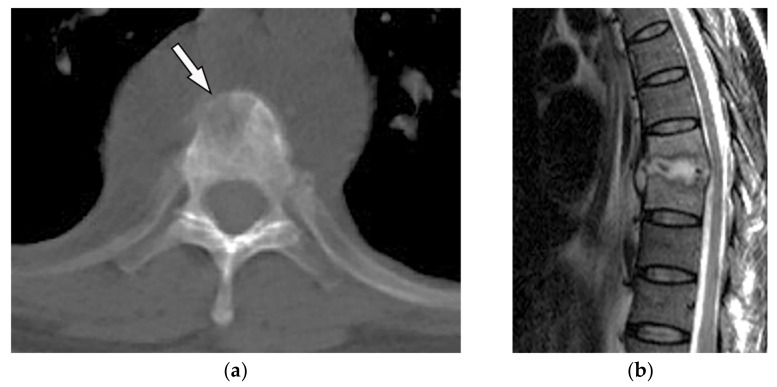
A 65-year-old man with fever, weight loss, and night sweeting. The non-contrast-enhanced CT (bone window) in axial plane (**a**) shows paraspinal soft tissue mass with erosion of right lateral aspect of adjacent vertebral body. Sagittal T2W image of another patient with the same pathology (**b**) shows hypersignal intensity within T8-T9 vertebral bodies with also intervertebral disc destruction and narrowing of spinal canal pushing the spinal cord posteriorly. Axial T1WFS + C (**c**) identified the enhancing paraspinal mass with peripheral rim enhancement (arrow) in its left posteromedial side, which is suggestive of abscess formation. Culture of aspirated pus under guide of CT was compatible with tuberculosis infection.

**Figure 14 diagnostics-12-00301-f014:**
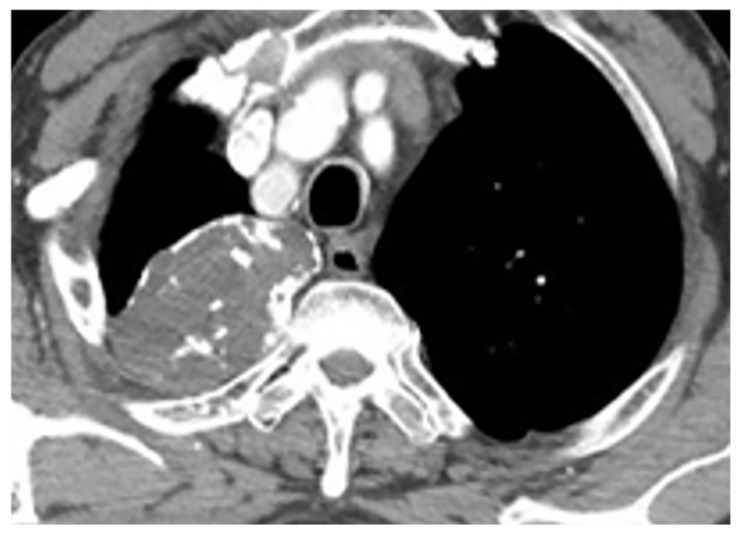
A 39-year-old woman with right vague chest pain. The contrast-enhanced axial CT demonstrates right-sided well-defined posterior mediastinal paraspinal mass with foci of calcifications within it. The round configuration is typical for peripheral nerve tumors. The attenuation is equal to chest wall muscles. Histopathological examination confirms Schwannoma.

**Figure 15 diagnostics-12-00301-f015:**
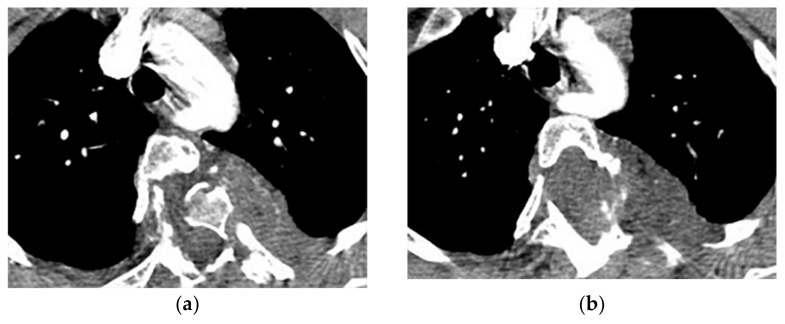
A 27-year-old male with scoliosis. Contrast-enhanced CT in the axial plane (soft tissue window) shows vertebral anomaly and dural ectasia. Left-sided well-defined lobular mass with the widening of adjacent neural foramen with intra- and extraspinal extension are offered (**a**–**c**), coronal CT (**d**) shows scoliosis in upper thoracic vertebrae, associated with two well-defined homogenous masses (arrow) with similar appearance in upper and lower posterior mediastinum.

**Figure 16 diagnostics-12-00301-f016:**
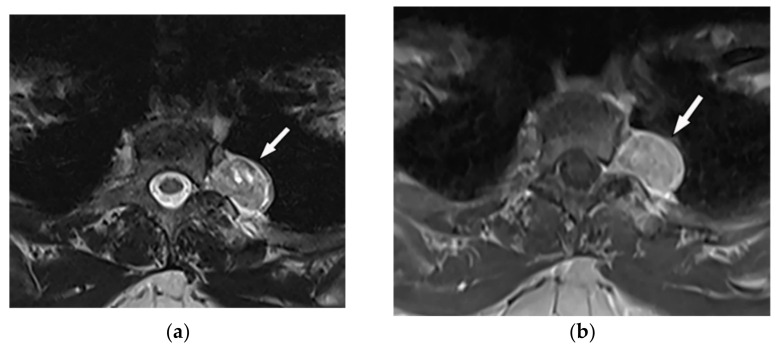
A 31-year-old woman with incidental findings. The axial T2W (**a**) and T1WFS + C (**b**) images show a sharply marginated lobular mass with heterogeneous peripheral hyper signal intensity in T2W, and internal hyper intensity of cystic changes, extending through the right-sided neural foramina (arrow). It has heterogeneous enhancement after contrast administration (**b**); no adjacent vertebral body scalloping was found.

**Figure 17 diagnostics-12-00301-f017:**
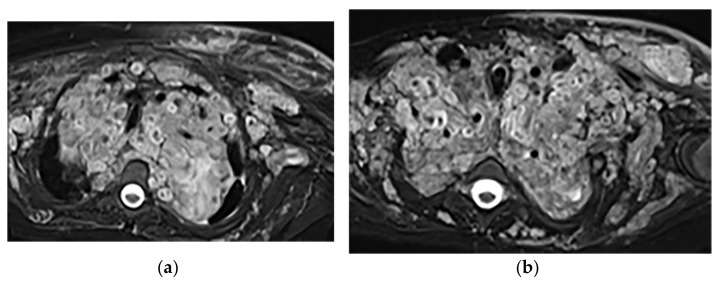
A 34-year-old man, with a case of neurofibromatosis type 1. The axial T2W images (**a**,**b**) show multiple confluent infiltrative paraspinal masses within the neck and upper thorax along the course of sympathetic chain and nerves with high signal intensity and central focus of hypointensity (target sign) surrounding the mediastinal vessels that are typical for plexiform Neurofibroma. Sagittal T2W (**c**) and coronal (**d**) images show the extensive infiltrative nature of this lesion with multi-compartment involvement and extension to pharyngeal space and pressure effect over cervical and thoracic vertebrae.

**Figure 18 diagnostics-12-00301-f018:**
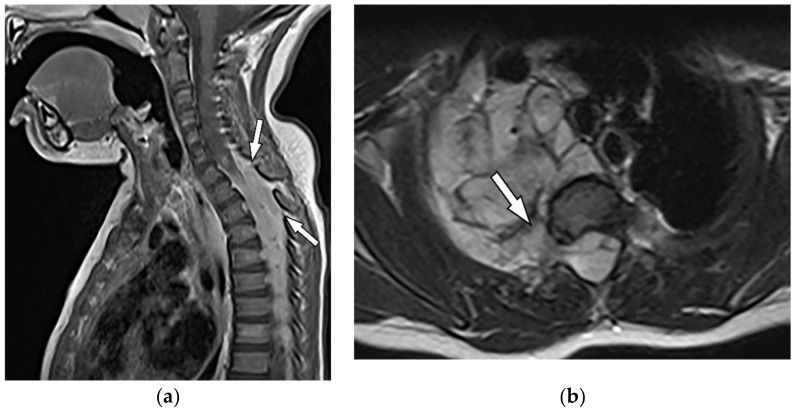
A 6-year-old boy with posterior mediastinal mass with histopathological confirmation for Neuroblastoma. Axial (**a**) T2W shows an ill-defined lobulated group (arrow) with heterogeneous and hyper-intense signal intensity and area of a signal void within the posterior mediastinum. It has intracanal extension via right-sided neural foramina and extradural components at multiple levels. It displaced the spinal cord anteriorly, as shown in sagittal T1W after gadolinium administration (arrow) (**b**).

**Figure 19 diagnostics-12-00301-f019:**
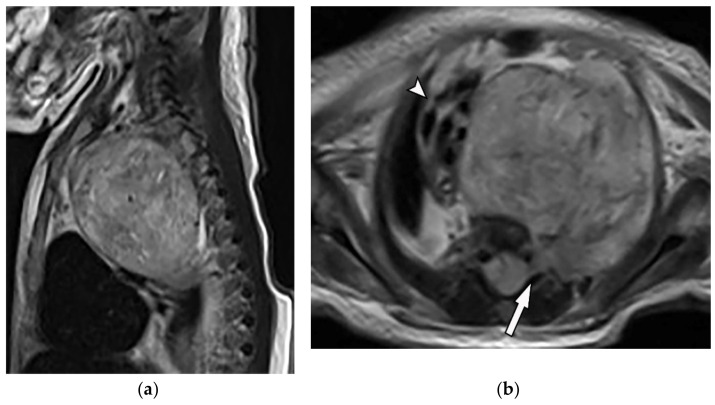
A 4-year-old girl with the opsomyoclonus-myoclonus syndrome. Sagittal (**a**) and axial (**b**) T2W images show large lobulated paraspinal masses (arrowhead) crossing the midline within the posterior superior mediastinum. It shows heterogeneous and hyperintense signal intensity with internal foci of the signal void caused by calcification. It extends through the spinal canal via neural foramina (arrow). Right anterolateral displacement of mediastinal great vessels is also identified.

**Figure 20 diagnostics-12-00301-f020:**
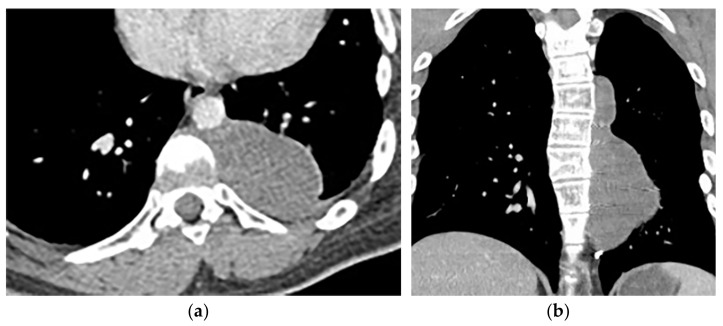
A 26-year-old man with incidental findings. The CT in axial and coronal views (**a**,**b**) shows a well-defined lobulated posterior mediastinal mass with homogenous attenuation and no significant enhancement after injection of contrast extending about five vertebral bodies length in the right posterior mediastinum with close contact to vertebral bodies, but with no vertebral erosion. Axial (**c**) and coronal T2W (**d**) of another patient with the same pathology show well-defined elliptical posterior mediastinal mass with heterogeneously high signal intensity intermixed with internal patchy and linear hypointensity. It also erodes the left lateral aspect of the vertebral body but with no intradural extension. Vertical orientation is typical for sympathetic chain tumors.

**Figure 21 diagnostics-12-00301-f021:**
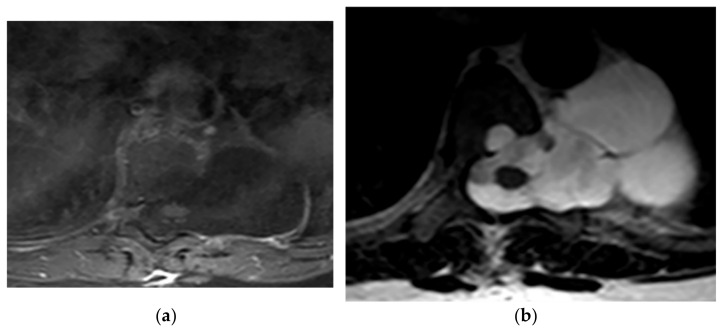
A 21-year-old man with Neurofibromatosis type 1. The axial T1FS (**a**) and T2W (**b**) show lateral expansion of CSF-filled sac through T6-T7 level with scalloping of the adjacent vertebral body. It has similar signal intensity to CSF in both T1W and T2W sequences and hypo and hypersignal intensity, respectively. No entrapped fat or neural elements were seen. Slightly anterior spinal cord displacement was also identified, which is compatible with lateral Meningocele.

**Figure 22 diagnostics-12-00301-f022:**
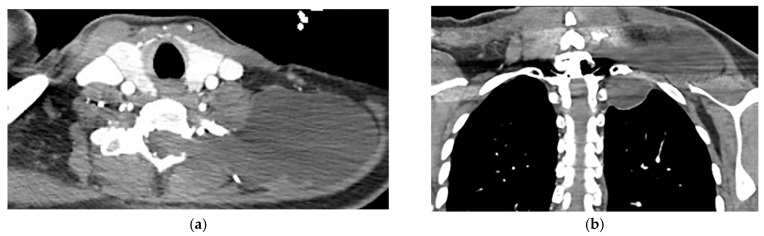
A 38-year-old man with a history of a remote motor vehicle collision. The axial (**a**) and coronal (**b**) contrast-enhanced CT show abnormal well-defined extraspinal fluid collection at the level C6-T1, which extends through the left neural foramina (lateral recess), communicating with CSF space. There is no edema, solid component, or abnormal enhancement within the mentioned collection or adjacent muscles. Regarding the history of trauma, a Pseudomeningocele diagnosis was made and was confirmed in MRI (not shown).

**Figure 23 diagnostics-12-00301-f023:**
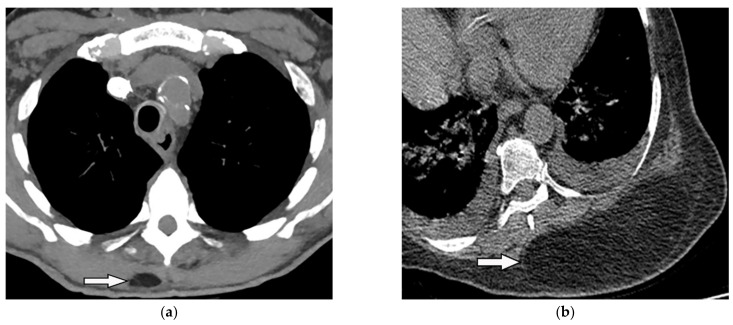
A 34-year-old man, with a case of SVC thrombosis with incidental finding. (**a**) Non-contrast-enhanced CT in axial plane demonstrates well-circumscribed lesion in the right posteromedial aspect of the chest wall with similar attenuation to subcutaneous fat with no internal septa. Multiple collaterals are also shown in the anterior aspect of the chest wall, maybe formed due to underlying SVC occlusion. (**b**) Axial plane CT of another patient shows large posteromedial chest wall mass with attenuation similar to adjacent subcutaneous fat compatible with lipoma.

**Figure 24 diagnostics-12-00301-f024:**
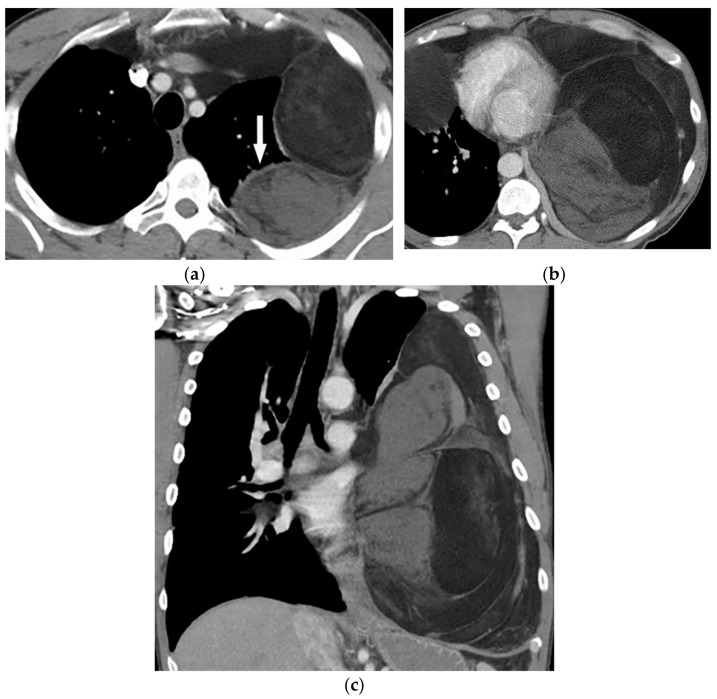
A 58-year-old man with dyspnea. (**a**,**b**) Axial contrast-enhanced CT shows a large heterogeneous mass with enhancing non-adipose solid components (arrow). The mass has extensive fat attenuation that is intermixed with soft tissue density. (**c**) Coronal image better characterizes the craniocaudal extension of the mass, which also shows a large inhomogeneous fat-containing lesion with an internal enhancing solid component.

**Figure 25 diagnostics-12-00301-f025:**
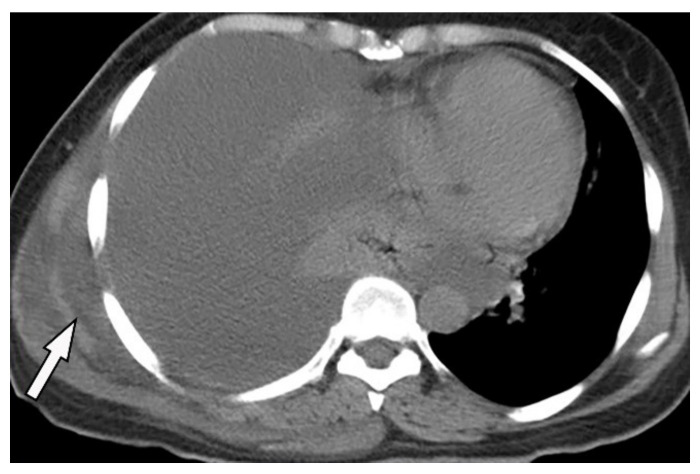
A 46-year-old man with high-grade fever and chills. The axial contrast-enhanced CT shows significant right-sided pleural effusion with the near complete collapse of the right lung resulting in a shift of the heart and mediastinum to the left side. There is pleural thickening and enhancement. There is an extrapleural component within the adjacent chest wall with rim enhancement (arrow). Aspiration was performed under the guidance of ultrasonography, and diagnosis of empyema necessitans was made as a complication of Actinomyces Israelii.

**Figure 26 diagnostics-12-00301-f026:**
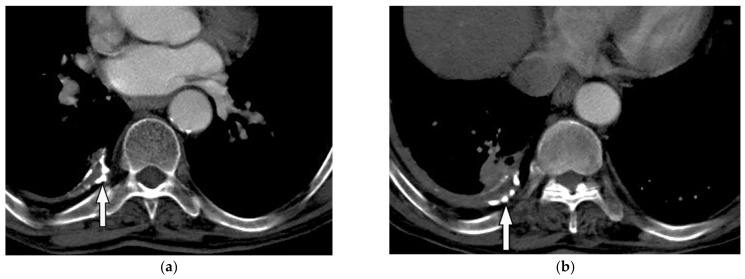
A 78-year-old man with dyspnea. The axial contrast-enhanced CT (**a**,**b**) demonstrates right-sided calcified pleural plaque (arrow) and small pleural effusion due to previous asbestosis exposure. Adjacent round atelectasis is also shown.

**Figure 27 diagnostics-12-00301-f027:**
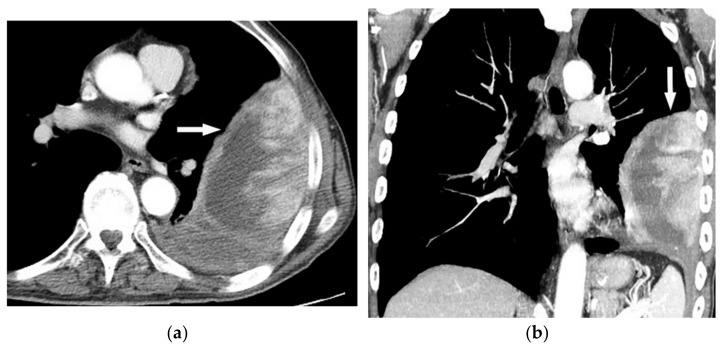
A 67-year-old man with dyspnea and chest pain. Contrast-enhanced CT in axial (**a**) and coronal (**b**) planes demonstrate left-sided localized enhancing pleural mass (arrow) with internal areas of necrosis that extend to the posteromedial aspect of the chest wall. Involvement of diaphragmatic pleura and elevation of left hemidiaphragm are also identified.

**Figure 28 diagnostics-12-00301-f028:**
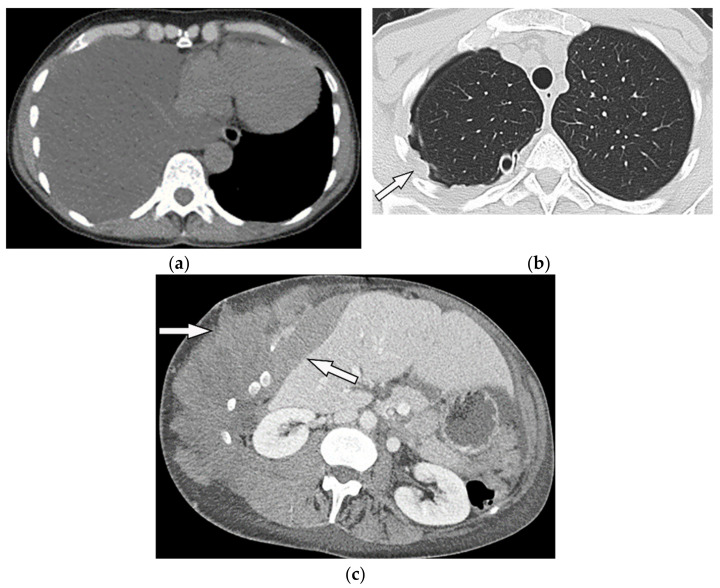
A 70-year-old man with dyspnea. The non-contrast CT in the axial plane demonstrates right-sided large pleural effusion, near complete collapse of the right lung (**a**) after the chest tube insertion; thick circumferential nodular pleural thickening of parietal pleura was shown (arrow) (**b**). Contrast-enhanced CT of the upper abdomen shows the extension of mesothelioma through the abdominal cavity and wall with indentation over adjacent liver parenchyma (arrow) (**c**).

**Figure 29 diagnostics-12-00301-f029:**
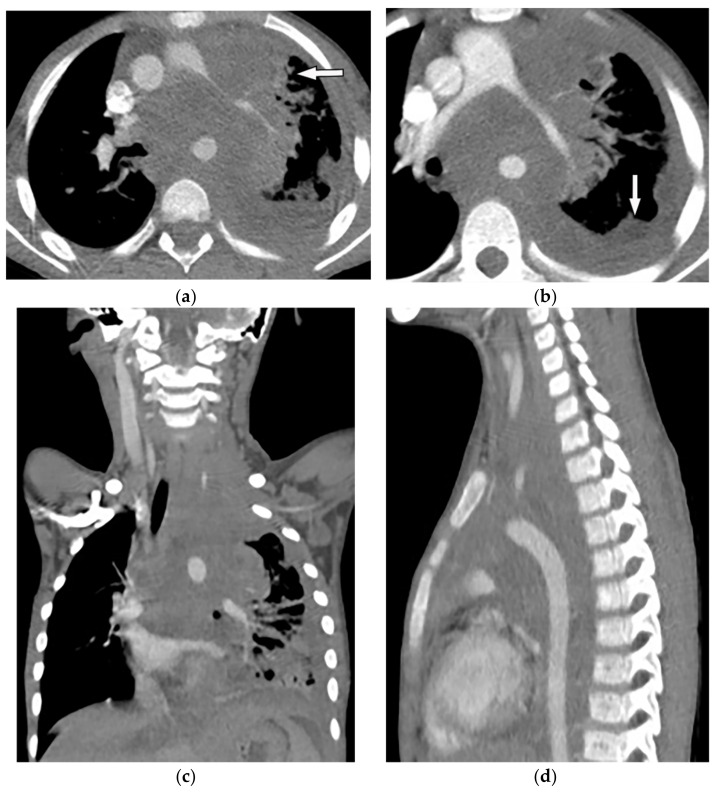
A 5-year-old girl with dyspnea. The axial contrast-enhanced CT in soft tissue window was obtained at the level of the main pulmonary artery (**a**,**b**), showing a large infiltrative soft tissue mass (arrow) with irregular border and slight enhancement encasing great mediastinal vessels. Left-sided pleural effusion was also identified. Coronal (**c**) and sagittal (**d**) CT better delineate the extension of the tumor and demonstrates infiltrative soft tissue mass within the superior, mid, and posterior mediastinum extending to the thoracic inlet and neck.

**Figure 30 diagnostics-12-00301-f030:**
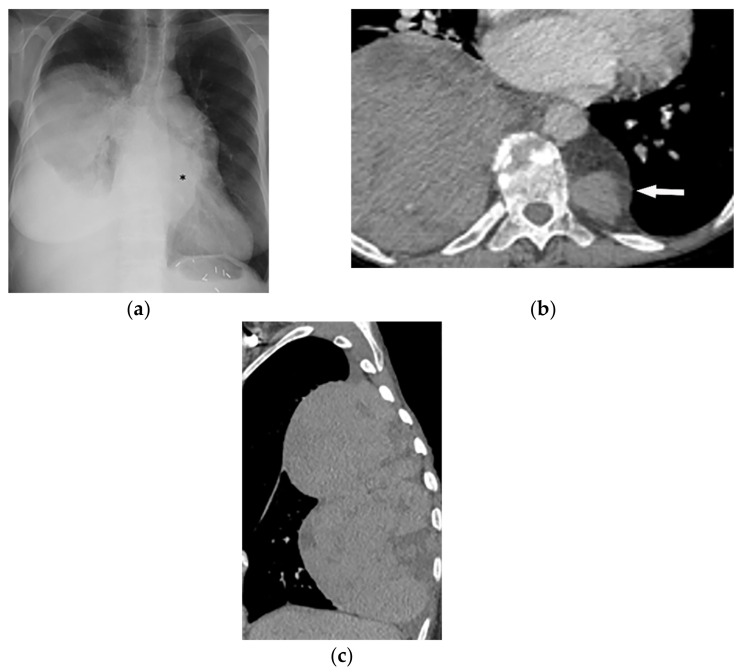
A 42-year-old woman with beta-thalassemia major presented with progressive dyspnea. PA radiograph (**a**) shows right basilar mass with meniscus sign and mass-like density over the mediastinum (asterisk). Surgical clips in LUQ of the abdomen from the previous splenectomy are noted. Axial CT with intravenous contrast (**b**) shows large heterogeneous well-defined soft tissue mass with some areas of internal hypo-density and no internal calcification in the right posteromedial chest wall; another similar appearing smaller lesion is detectable on the left side (arrow). Sagittal CT of the chest (**c**) along the long axis of the lesion demonstrates the craniocaudal extension of the lobulated retro-mediastinal mass. Biopsy confirmed the diagnosis of EMH.

**Table 1 diagnostics-12-00301-t001:** Posteromedial chest wall tumors classification based on site of origin.

Origin	Malignant Neoplasm	Benign Neoplasm
Osseous and cartilaginous lesions - Rib - Thoracic spine	Osteosarcoma: - Osseous osteosarcoma - Extraosseous osteosarcoma Chondrosarcoma Ewing sarcoma: - Ewing sarcoma of bone - Extraosseous Ewing sarcoma Bone lymphoma Askin tumor Multiple myeloma Solitary plasmocytoma of bone	Osteochondroma Aneurysmal bone cyst Fibrous dysplasia of bone Ossifying fibromyxoid tumor Giant cell tumor Chondromyxoid fibroma Enchondroma Langerhans cell histiocytosis
Vascular lesions - Aorta - Intercostal vessels	Angiosarcoma Kaposi sarcoma	Lymphangioma Hemangioma
Adipose tumors	Liposarcoma	Lipoma Spindle cell lipoma
Neurogenic tumors - Intercostal nerve - Sympathetic nerve	Malignant peripheral nerve sheet tumor Neuroblastoma	Schwannoma Neurofibroma Ganglioneuroma Paraganglioma Meningocele
Lung and pleural lesions invading the chest wall	Mesothelioma Drop metastasis of thymic malignancy Lung malignancy with chest wall invasion	Localized fibrous tumor of pleura Empyema necessitans
Cutaneous lesions	Dermatofibrosarcoma protuberance	Cavernous hemangioma Epidermal inclusion cyst
Fibrous and muscle tumors	Undifferentiated pleomorphic sarcoma	Fibromatosis
Miscellaneous tumors	Neurofibrosarcoma	Extramedullary hematopoiesis Castleman disease Monoclonal immunoglobulin deposition diseases (MIDDs)
Secondary tumors	Bone metastasis	NA

**Table 2 diagnostics-12-00301-t002:** Imaging manifestations of posteromedial chest wall osseous lesions.

Tumor Type	Imaging Findings	
	CT	MRI
**Malignant**		
Osteosarcoma	Dense central calcification, expansile bone remodeling, periosteal reaction, pathologic fracture Sparing of the intervertebral disk while decreasing the height of the vertebral body Lung and nodal metastasis	T1W: low to intermediate signal (high signal in hemorrhage) T2W: hyperintense Mineralization: hypointense on both T1W and T2W T1W FS + C: heterogeneous enhancement
Ewing sarcoma	Lytic bone destruction with ill-defined border, rare calcification, heterogenous paraspinal soft tissue with soft tissue larger than osseous component	T1W: iso to hyperintense to the muscle (high signal in hemorrhage) T2W: heterogeneous to hyperintense T1W FS + C: intense homogenous or heterogeneous enhancement
Chondrosarcoma	Well-defined mass: soft tissue+ mineralization Calcification: rings and arcs, stippled or dense Invasion of adjacent structures	T1W: variable, iso to hypointense to muscle T2W: overall hyperintense Mineralization: hypointense on both T1W and T2W T1WFS + C: heterogeneous enhancement
Multiple Myeloma	Osteolytic lesion with endosteal scalloping, diffuse osteopenia, multiple small lesions with mottled appearance, and osteoporotic fracture	Five patterns of marrow involvement: normal, focal, diffuse, combined diffuse and focal pattern, salt and pepper appearance. T1W: hypointense (significantly in the later phase of the disease) T2W: intermediate to hyperintense DWI: hyperintense T1WFS + C: enhancement expected
Solitary Plasmacytoma of the Bone	Extrapleural mass with well-circumscribed margin and “soap bubble” appearance and rare calcification, multicystic expansion	T1W: hypointense T2W: hyperintense
**Benign**		
Aneurysmal bone cyst	Well-defined expansile osteolytic lesion with thin marginal sclerosis and typical fluid-fluid level with internal septation	Fluid-fluid level T1W: hyperintense secondary to subacute age of internal hemorrhage T1W FS + C: can be seen in solid component of secondary ABCs
Fibrous dysplasia	Well-defined intramedullary osteolytic lesion with fusiform bony expansion and endosteal scalloping with preservation of cortical contour, sclerotic margin, trabeculation, and cortical thickening, “Ground glass” appearance, sometimes completely radiolucent or sclerotic	T1W: hypointense T2W: variable low to high signal depending on varying amounts of fibrous tissue
Giant cell tumor	Osteolytic lesion with bone expansion, cortical thinning, and heterogeneous soft-tissue attenuation with area of hemorrhage or necrosis	T1W, T2W: low to intermediate intensity representative of the abundant internal amount of hemosiderin and collagen
Enchondroma	Focally expansile well-demarcated osteolytic lesion, with or without cortical bulging, matrix calcification	T1W: significantly hypointense T2W: significantly hyperintense T1WFS + C: contrast uptake is uncommon unless in the small enchondromas or peripheral enhancement. Internal calcification: hypointense in all sequences
Chondromyxoid fibroma	Cortical expansion with lobulated border, abundant peripheral sclerosis, and rarely internal matrix calcification	T1W: isointense with hypointense rim T2W: intermediate to significantly hyperintense with hypointense rim T1WFS + C: diffuse moderate to intense enhancement
Chondroblastoma	Oval or round well-circumscribed lesion with internal mineralization and variable aggressiveness. Radiography is more accurate than MRI for diagnosis.	T1W: homogenously hypointense T2W: heterogeneously signal intensity with common peri-tumoral marrow edema, ABC changes, and periosteal reaction resembling malignant bone lesions.
Paget’s disease of the rib	Osseous expansion, cortical thickening, and trabecular coarsening	Blastic phase: hypointense on both T1W and T2W images Lytic phase: speckled hypointense on T1W and hyperintense on T2W, T1WFS + C enhancement

T1W = T1-weighted, T2W = T2-weighted, FS = fat saturated, C = contrast; DWI = Diffusion weighted imaging.

**Table 3 diagnostics-12-00301-t003:** Imaging manifestations of posteromedial chest wall soft-tissue tumors.

Tumor Type	Imaging Findings	
	CT	MRI
**Primary Neurogenic Tumors**		
Schwannoma	Well-defined mass with homogenous attenuation, “fat-split” sign, internal calcification in long-standing schwannomas, postcontrast enhancement except for areas of necrosis.	T1W: iso or slightly hyperintense; T2W: significantly hyperintense
Neurofibroma	Well-circumscribed mass with smooth margin and soft tissue attenuation, possible internal calcifications, rib erosion, neural foramina widening because of tumor extension along with the spinal nerve roots.	T2W, T1WFS + C: so-called “target sign” appearance: hyperintense rim and hypointense center
Neuroblastoma [7,16,43]	Ill-defined paravertebral soft tissue mass with heterogeneous attenuation with internal calcification in at least 30% of cases (spotty calcification).	T1W: hyposignal T2W: hyperintense T1WFS + C: heterogeneous enhancement Calcification has a signal void in all sequences
Ganglioneuroma [7,16,43]	Homogenous or heterogeneous attenuation with internal calcification in 25% of cases.	T1W, T2W: intermediate signal with the curvilinear or nodular low signal band making the whorled appearance
**Lipomatosis Tumors**		
Lipoma	Homogenous similar attenuation to macroscopic fat with approximate HU: −100.	T1W, T2W: signal intensity identical to subcutaneous fat T1WFS + C: no enhancement (mild enhancement can be visible for septa < 2 mm thickness)
Liposarcoma	A heterogeneous mass mixture of fat and soft tissue: higher attenuation than normal fat (hypercellularity), necrosis, and calcification in myxoid subtype. Attenuation similar to fat in well-differentiated subtype. Thick septa, enhancing solid component.	T1W: variable hyperintense (myxoid liposarcoma), hypointense (well-differentiated), and intermixed hyper and hypointense (dedifferentiated subtype) T2W: hyperintense (myxoid liposarcoma and dedifferentiated subtype) T1WFS + C: variable enhancement
**Others**		
Rhabdomyosarcoma	Invasive, destructive homogenous mass with no mineralization and rapid growth with adjacent soft tissue and bone invasion.	T1W: isointense T2W: hyperintense with hypointense areas reflecting area of necrosis (alveolar and pleomorphic subtypes) T1WFS + C: homogenous or ring-like enhancement
Mesothelioma	Circumferential pleural thickening, bony or cartilaginous differentiation, unilateral pleural effusion, interlobular septal thickening, tumoral extension, thoracic and extrathoracic metastasis.	T1W: unilateral hyperintense pleural effusion, iso to slightly hyperintense pleural thickening T2W: moderately hyperintense T1WFS + C: typical enhancement is expected
Extramedullary Plasmacytoma	Soft tissue masses with nonspecific imaging manifestations. Larger lesions show aggressive behavior such as infiltration, destruction, and encasement.	T1W: isointense T2W: iso to hyperintense T1WFS + C: variable enhancement (from mild to marked enhancement)

T1W = T1-weighted, T2W = T2-weighted, FS = fat saturated, C = contrast; DWI = Diffusion weighted imaging.

**Table 4 diagnostics-12-00301-t004:** Imaging manifestations of posteromedial chest wall soft-tissue tumor-like lesions.

Tumor Type	Imaging Findings	
	CT	MRI
**Neurogenic**		
Lateral Meningocele	Well-circumscribed paravertebral mass with attenuation similar to CSF. CT myelography: ipsilateral neural foramina enlargement communicating with subarachnoid space.	T1W: hypointense T2W: hyperintense (similar intensity to CSF) T1WFS + C: lack of enhancement
Pseudomeningocele	Differentiated from Meningocele by lack of dura wrapping.	T1W: hypointense T2W: hyperintense (similar intensity to CSF) T1WFS + C: lack of enhancement [47]
**Others**		
Extramedullary hematopoiesis	Heterogeneous mass with internal foci of fat with lack of calcification.	T1W, T2W: heterogeneous with internal foci of hyperintensity in old lesions (representative of fat), the intermediate intensity with subtle or no enhancement in active lesions
Asbestos-related pleural plaques	Calcified or non-calcified focal pleural thickening, “Comet tail” appearance usually seen in lower lobes [48].	T1W: hypo to isointense T2W: hypointense (due to fibrosis or calcification) [49]
Empyema necessitance	Connection of pleural collection to extrapleural mass, soft tissue inflammation, rib destruction with periosteal reaction, and fluid collection.	T1W: hypointense effusion and fluid collection T2W: hyperintense effusion, increased thickness of extrapleural fat, and chest wall muscles with hyperintense on T2WFS T1WFS + C: pleural and septal enhancement

T1W = T1-weighted, T2W = T2-weighted, FS = fat saturated, C = contrast; DWI = Diffusion weighted imaging.

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
