# Peer review of "A Review of Posteromedial Lesions of the Chest Wall: What Should a Chest Radiologist Know?"

_diagnostics, 2022, doi:10.3390/diagnostics12020301_

Round 1
Reviewer 1 Report
The manuscript represents a review of the posteromedial lesions of the chest wall with focus on the radiological characteristics, that is in my opinion correctly conducted and comprehensive.
Figures depicting every lesion described are an added value in comparison with previous works.
A few specific remarks:
- In the abstract it seems that this sentence is not complete “A systematic approach based on age, clinical history, radiologic findings, and is required for correct diagnosis.” (lines 20-21, page 1).
- I don’t think reference [6] in line 39 is appropriate to the previous sentence “Neurogenic tumors are more commonly arising from posteromedial chest wall rather than other sites in the chest wall” (lines 38-39, page 1) as the article does not mention neurogenic tumors incidence in the posteromedial chest wall compared to other sites of the chest wall but presents the anatomy of the region. Could you specify why you inserted the reference after this specific sentence?
- Figure 7, page 13: it seems that image (B) and (C) descriptions are inverted.
- Paragraph 9.1.3 line 405, page 24, did you mean “neuroblastoma” instead of “neurofibroma”?
Author Response
- The manuscript represents a review of the posteromedial lesions of the chest wall with focus on the radiological characteristics, that is in my opinion correctly conducted and comprehensive. Figures depicting every lesion described are an added value in comparison with previous works.
- In the abstract it seems that this sentence is not complete “A systematic approach based on age, clinical history, radiologic findings, and is required for correct diagnosis.” (lines20-21, page1).
- Authors’ response and Action:
Thank you for this comment. We agree with the Reviewer’s comment. We have reviewed our manuscript and changed the sentence as “A systematic approach based on age, clinical history and radiologic findings is required for correct diagnosis”.
- I don’t think reference [6] in line 39 is appropriate to the previous sentence “Neurogenic tumors are more commonly arising from posteromedial chest wall rather than other sites in the chest wall” (lines 38-39, page 1) as the article does not mention neurogenic tumors incidence in the posteromedial chest wall compared to other sites of the chest wall but presents the anatomy of the region. Could you specify why you inserted the reference after this specific sentence?
- Authors’ response and Action:
Thank you for this comment. We agree with the Reviewer comment. We have reviewed our manuscript and changed the sentence as “Neurogenic tumors are more commonly arising from the posteromedial chest wall as they originate from autonomic ganglia, paraganglia, or nerve sheets. So they account for the majority of lesions found in the posterior mediastinum and chest wall”. The reference was also corrected and changed to Pavlus JD, Carter BW, Tolley MD, Keung ES, Khorashadi L, Lichtenberger JP, 3rd. Imaging of Thoracic Neurogenic Tumors. AJR Am J Roentgenol. 2016;207(3):552-61.
- Figure 7, page 13: it seems that image (B) and (C) descriptions are inverted.
- Authors’ response and Action:
Thank you for this comment. We agree with the Reviewer comment. We have reviewed our manuscript and changed the descriptions order.
- Paragraph 9.1.3 line 405, page 24, did you mean “neuroblastoma” instead of “neurofibroma”?
- Authors’ response and Action:
Thank you for this comment. We agree with the Reviewer comment. We have reviewed our manuscript and changed “neurofibroma” to “neuroblastoma”.

Reviewer 2 Report
Review Comments
The authors explained a comprehensive review of Poster medial Lesions of the Chest Wall. The presented paper explained the posteromedial segment of the thoracic wall. However, the following major corrections can be considered by the authors to further improve the quality of the manuscript.
I have some major corrections and suggestions as below:-
- The abstract can be improved. The outcome of the review work in terms of complete review details must be included in the abstract.
- The novel contribution of the paper should be included. For readers to quickly catch your contribution, it would be better to highlight major difficulties and challenges, and your original achievements to overcome them, in a clearer way in the abstract and introduction. The organization of the paper should be added at the end of the introduction.
- The introduction section is not sufficient and some more motivation related to topics should be included in the introduction section?
- Future work and limitations of the various reviews must be included.
- In Section 4 Role of Imaging, Incomplete details are available like “Bone Lesions:”. Various types of imaging techniques can be discussed as tree graphs.
- Statics related to various published papers is the presented review work is missing. Authors must provide a brief overview of papers published in various journals related to review work.
- Conclusion sections are presented very short. Authors must conclude all section that has been discussed in the review work.
- Authors must add a few tables with proper citations at various sections related to various literature in terms of their work, achieved outcomes, remarks, and various other required parameters.
- The research direction of proposed work for upcoming researchers must be added and discussed briefly.
- Resolutions and proper naming of all figures must be addressed.
- There are various typos and grammatical must be addressed. There are some occasional grammatical problems within the text. It may need the attention of someone fluent in the English language to enhance the readability.
- Acronyms of various abbreviations must be included.
- Definitions of Various performance parameters must be discussed.
- Details of various data set sources for imaging finding requirements must be added as a separate section with proper tabulated information and citations.
- Some more recent papers can be added and discussed. Moreover, the manuscript could be substantially improved by relying and citing more on recent literature about contemporary real-life case studies of soft computing techniques in different fields.
- Authors utilize and cite the paper sequentially from the references.
- Work-related to various expert systems an AI/ML must be included.
- Authors must discuss the various hardware and software requirements.
- What are the other feasible alternatives? What are the advantages of adopting this technique over others in this case? How will this affect the results? More details should be furnished.
- What are the other feasible alternatives? What are the advantages of adopting this technique over others in this case? How will this affect the results? More details should be furnished.

Author Response
- The authors explained a comprehensive review of Poster medial Lesions of the Chest Wall. The presented paper explained the posteromedial segment of the thoracic wall. However, the following major corrections can be considered by the authors to further improve the quality of the manuscript. I have some major corrections and suggestions as below: -
- The abstract can be improved. The outcome of the review work in terms of complete review details must be included in the abstract.
- Authors’ response and Action:
Thank you for this comment. We agree with the Reviewer comment. We have reviewed the abstract and added some sentences in order to be improved.
The revised abstract is:
“A heterogeneous group of tumors can affect the posteromedial chest wall. They form diverse groups of benign and malignant (primary or secondary) pathologies that can arise from different chest wall structures, i.e., fat, muscular, vascular, osseous, or neurogenic tissues. Chest radiography is very nonspecific for the characterization of chest wall lesions. The modality of choice for the initial assessment of the chest wall lesions is computed tomography (CT). More advanced cross-sectional modalities such as magnetic resonance imaging (MRI) and positron emission tomography (PET) with fluorodeoxyglucose are usually used for further characterization, staging, treatment response, and assessment of recurrence. A systematic approach based on age, clinical history, radiologic findings, and is required for correct diagnosis. It is important for radiologists to be familiar with the spectrum of lesions that might affect the posteromedial chest wall and their characteristic imaging features. A systematic approach based on age, clinical history, and radio-logic findings is required for correct diagnosis. It is essential for radiologists to be familiar with the spectrum of lesions that might affect the posteromedial chest wall and their characteristic imaging features. Although the imaging findings of these tumors can be nonspecific, cross-sectional im-aging helps to limit the differential diagnosis and determine the further diagnostic investigation (e.g., image-guided biopsy). Specific imaging findings, e.g., location, mineralization, enhancement pattern, and local invasion, occasionally allow a particular diagnosis.
This article reviews the posteromedial chest wall anatomy and different pathologies. We provide a combination of location and imaging features of each pathology. We will also explore the role of imaging and its strengths and limitations for diagnosing posteromedial chest wall lesions.”
- The novel contribution of the paper should be included. For readers to quickly catch your contribution, it would be better to highlight major difficulties and challenges, and your original achievements to overcome them, in a clearer way in the abstract and introduction. The organization of the paper should be added at the end of the introduction.
- Authors’ response and Action:
Thank you for this comment. We agree with the Reviewer comment. We have reviewed the abstract and introduction and improved the content.
- The introduction section is not sufficient and some more motivation related to topics should be included in the introduction section?
- Authors’ response and Action:
Thank you for this comment. We agree with the Reviewer comment. We have reviewed the introduction and added some sentences in order to be improved.
Revised text is as follow:
“Introduction:
Chest wall tumors are uncommon causes of thoracic neoplasms, which are less common than soft tissue or bony neoplasms elsewhere. Unfamiliarity with the complex posteromedial chest wall anatomy and radiologic features of related neoplasms is a diagnostic dilemma for radiologists. These tumors are heterogeneous with nonspecific clinical manifestations and different imaging characteristics, which make their diagnosis challenging. Either a benign or malignant nature and primary or secondary origin are probable. Primary chest wall neoplasms originate from chest wall structures, e.g., bony thorax, cartilage, muscle, fat, blood vessels, and nerve sheet. Secondary chest wall neoplasms include direct invasion from adjacent malignancies (lung or breast carcinomas) or distant metastasis.
The posteromedial aspect of the chest wall has complex anatomy due to the presence of intercostal nerves, sympathetic chain, and vascular structures. Many neoplasms originate from these structures. Some of them may be almost exclusive to this location. Neurogenic tumors are more commonly arising from the posteromedial chest wall as they originate from autonomic ganglia, paraganglia, or nerve sheets. So, they account for the majority of lesions found in the posterior mediastinum and chest wall. Many of these lesions have specific imaging characteristics that help make precise diagnoses and avoid invasive sampling. In other conditions with nonspecific imaging appearance, cross-sectional imaging plays an essential role in limiting the differential diagnosis and defining the further investigation, e.g., imaging-guided biopsy. So, it is crucial for radiologists to be familiar with these diverse group of lesions and their imaging characteristics.
Previous studies mostly focused on the assessment of malignant lesions of the chest wall. None of them specifically evaluated the lesions of the posteromedial chest wall. Only one review article investigated the paravertebral masses in the thoracic boundary. This study categorized lesions into neurogenic tumors, non-neurologic tumors, and non-neoplastic masses. To the best of our knowledge, our review is the only one focusing on the posteromedial aspect of the chest wall, addressing nearly all of the lesions that could be found in this anatomic location. This article reviews the posteromedial chest wall anatomy and different pathologies. We illustrated the imaging features of each lesion, e.g., the location, presence of calcification, adjacent bone destruction, the pattern of enhancement, and appearance on magnetic resonance imaging and positron emission tomography. We also explored the role of imaging and its strengths and limitations for diagnosing posteromedial chest wall lesions.”
- Future work and limitations of the various reviews must be included.
- Authors’ response and Action:
Thank you for this comment. We agree with the Reviewer’s comment and it is addressed at the end of conclusion.
Added text is as follow:
“Future investigation is required using quantitative novel imaging modalities to increase the diagnostic accuracy of radiology. Furthermore, improvement in deep learning and radiomics may increase patients' benefit from reduced need for biopsy and individualized treatment options.”
- In Section 4 Role of Imaging, Incomplete details are available like “Bone Lesions:”. Various types of imaging techniques can be discussed as tree graphs.
- Authors’ response and Action:
Thank you for this comment. We agree with the Reviewer comment. The “bone lesions” was the “topic” of next malignant bone lesions that we removed in the revised version.
The graph was added as below:
- Statics related to various published papers is the presented review work is missing. Authors must provide a brief overview of papers published in various journals related to review work.
- Authors’ response and Action:
Thank you for this comment. We agree with the Reviewer’s comment. A paragraph assessing the previous studies was added to introduction.
Added paragraph is:
“Previous studies mostly focused on the assessment of malignant lesions of the chest wall. None of them specifically evaluated the lesions of the posteromedial chest wall. Only one review article investigated the paravertebral masses in the thoracic boundary. This study categorized lesions into neurogenic tumors, non-neurologic tumors, and non-neoplastic masses. To the best of our knowledge, our review is the only one focusing on the posteromedial aspect of the chest wall, addressing nearly all of the lesions that could be found in this anatomic location.”
- Conclusion sections are presented very short. Authors must conclude all section that has been discussed in the review work.
- Authors’ response and Action:
Thank you for this comment. We agree with the Reviewer comment. We have reviewed the conclusion part and added some sentences in order to be improved.
The revised conclusion is:
“Chest wall neoplasms are a group of heterogeneous lesions, and the posteromedial chest wall is a source of different pathologies due to its complex anatomy. Many of these pathologies can be differentiated by imaging. A comprehensive systematic approach with varying imaging modalities is needed to identify the correct diagnosis or limit the differential diagnosis and determine the appropriate further investigation. This article illustrates the various posteromedial chest wall pathologies and their imaging features. Ill-defined border, heterogeneous enhancement, and local invasion are more suggestive of a malignant lesion. In contrast, well-defined borders and the absence of local invasion or distant metastasis favor benign nature. The pattern of mineralization helps in the differentiation of osseous/cartilaginous neoplasm from other neoplasms. We also explored key imaging features as well as strengths and limitations of each imaging modality. Therefore, the familiarity of radiologists with the imaging features of posteromedial chest wall lesions is crucial and can avoid unnecessary invasive procedures. Future investigation is required using quantitative novel imaging modalities to increase the diagnostic accuracy of radiology. Furthermore, improvement in deep learning and radiomics may increase patients' benefit from reduced need for biopsy and individualized treatment options.”
- Authors must add a few tables with proper citations at various sections related to various literature in terms of their work, achieved outcomes, remarks, and various other required parameters.
- Authors’ response and Action:
Thank you for this comment. We agree with the Reviewer comment. We have added citations in the tables.
- The research direction of proposed work for upcoming researchers must be added and discussed briefly.
- Authors’ response and Action:
Thank you for this comment. We agree with the Reviewer’s comment. We have addressed this comment in the revised introduction.
- Resolutions and proper naming of all figures must be addressed.
- Authors’ response and Action:
Thank you for this comment. We have reviewed and revised all the included figures and figure legends.
- There are various typos and grammatical must be addressed. There are some occasional grammatical problems within the text. It may need the attention of someone fluent in the English language to enhance the readability.
- Authors’ response and Action:
Thank you for this comment. We have reviewed the text and addressed all the typos and grammatical points.
- Acronyms of various abbreviations must be included.
- Authors’ response and Action:
Thank you for this comment. We added abbreviations.
Abbreviations added as follow:
“Abbreviations: MRI= Magnetic resonance imaging, CT= Computed tomography, 18F-FDG= Fluorine 18 fluorodeoxyglucose, PET/CT= Positron emission tomography/computed tomography, GCT= Giant cell tumor, MM= Multiple myeloma, SBP= Solitary plasmacytoma of the bone, ABC= Aneurysmal bone cyst, FD= Fibrous dysplasia, DCE= Dynamic contrast-enhanced”
- Definitions of Various performance parameters must be discussed.
- Authors’ response and Action:
I would appreciate it if you could be more specific about various performance parameters and explain more about it so we could address your comment. Performance analysis is not within the scope of this text. Authors have used imaging modality of choice to depict pathologies and discuss diagnostic importance of such pathologies.
- Details of various data set sources for imaging finding requirements must be added as a separate section with proper tabulated information and citations.
- Authors’ response and Action:
Thank you for this comment. Included images has obtained basically from different imaging centers with different protocols so we are not able to provide single imaging protocol.
- Some more recent papers can be added and discussed. Moreover, the manuscript could be substantially improved by relying and citing more on recent literature about contemporary real-life case studies of soft computing techniques in different fields.
- Authors’ response and Action:
Thank you for this comment. We agree with editor comment, and we added some new and updated references.
- Authors utilize and cite the paper sequentially from the references.
- Authors’ response and Action:
Thank you for this comment. We agree with the Reviewer comment. We have reviewed the manuscript and reordered the references.
- Work-related to various expert systems an AI/ML must be included.
- Authors’ response and Action:
Thank you for this comment. A paragraph about AI and ML was added.
The added paragraph is:
On the other hand, recent advances in deep learning and artificial intelligence (AI) provide the ability of automatic classification, disease detection, and segmentation. Chest radiography and CT scan are excellent candidates for developing deep learning algorithms. AI has the potential to detect visual information and perform quantitative analyses. Besides, radiomics can be used to characterize the benign or malignant nature of a lesion, predict the prognosis and the probability of response to treatment of the malignant lesions.
- Authors must discuss the various hardware and software requirements.
- Authors’ response and Action:
Thank you for this comment. Various hardware and software were used so we are not able to provide more detailed and uniform information to explain all the used software and hardware. In addition, this is beyond the scope of this text, which is focused solely on diagnostic interpretation of findings and not technical aspects behind such diagnosis.
- What are the other feasible alternatives? What are the advantages of adopting this technique over others in this case? How will this affect the results? More details should be furnished.
- Authors’ response and Action:
Thank you for this comment. Diving into alternative means of diagnosis is beyond the scope of this text. We have used the imaging modality of choice to show representative imaging for each presented pathology. We believe that explaining all alternative diagnostic tools would make the text long and most likely less desirable for the readers of this article.

Round 2
Reviewer 2 Report
The revised version has been improved by authors who replied to my questions and took into account all my comments during the revision process.
The current version is acceptable to be published.